# Psychological interventions that decrease psychological distance or challenge system justification increase motivation to exert effort to mitigate climate change

Jo Cutler [1,2,3,14] ✉, Luis Sebastian Contreras-Huerta [1,2,3,4,5,14], Boryana Todorova [6], Jonas Nitschke [6], Katerina Michalaki [7,8], Lina Koppel[9], Theofilos Gkinopoulos[10,13], Todd A. Vogel [1,2,3], Claus Lamm [6], Daniel Västfjäll [11], Manos Tsakiris [7,8], Matthew A. J. Apps [1,2,3,15] & Patricia L. Lockwood [1,2,4,12,15] ✉

Climate change is one of the biggest challenges facing humanity. To limit its damaging impacts, billions of people must take pro-environmental actions. However, these often require effort and people avoid effort. It is vital to identify psychological interventions that increase willingness to exert effort. 3055 people from six diverse countries completed an effort-based decision-making task (Pro-Environmental Effort Task; Bulgaria: n = 404, Greece: n = 85, Nigeria: n = 660, Sweden: n = 1090, UK: n = 482, USA: n = 334). Participants chose whether to exert physical effort (50-95% of their maximum) to reduce carbon emissions, after experiencing one of 11 expert crowd-sourced interventions or no intervention. We applied computational modelling to precisely quantify motivation to help the climate, compared to a closely matched non-environmental cause. We found two interventions, which reduced the psychological distance to climate change impacts or promoted climate action as patriotic and protecting participants' way of life, had consistent positive effects on increasing effortful pro-environmental behaviours, across measures and control analyses. At the individual level, motivation to benefit the climate was associated with belief in climate change and support for pro-environmental policies. In contrast, trait apathy and effort aversion were linked with reduced motivation to benefit both the climate and food cause. Together, our results have crucial implications for promoting effortful actions that help mitigate climate change.

Climate change is having devastating and worsening impacts on people, wildlife, flora, and societies[1,2]. To reduce these negative impacts, billions of people will need to change their behaviours and choose actions that benefit the environment. Psychological interventions may offer highly fruitful tools to promote pro-environmental action. Evidence shows that they increase pro-climate attitudes or intentions, with some consistency across diverse samples globally[3–5]. Interventions have also effectively increased behaviour in field studies[6–8]. However, these approaches rarely test whether any intervention effects are specific to climate behaviours. Such methods also cannot identify the mechanisms of pro-environmental decision-making: how choices integrate the environmental benefits with the required costs. This is particularly pertinent in the context of climate change, where many

behaviours require physical effort. We often need to decide between a more physically effortful option with greater environmental benefit or a less effortful option that is worse for the environment. For example, we choose to walk or drive, to repair or replace items, and whether to clean and sort waste for recycling. Effective psychological interventions must change how willing people are to exert physical effort.

One major challenge is that humans and other animals generally find effort aversive and avoid it, even when exerting effort obtains rewards, known as 'the theory of least effort' or effort aversion[9–14]. This aversion to effort is magnified in social contexts where the direct benefit is not immediately for ourselves, with people less willing to engage in effortful behaviours that help others[15–17]. Effort aversion may therefore be a significant

barrier to actions that reduce climate change, which often do not have immediate benefits for the individual[6]. Other suggested theoretical barriers to pro-environmental actions are that the consequences of climate change are psychologically distant: uncertain, far in the future, affecting distant places, and impacting people different to themselves[18,19]. While some of these factors may also apply to other global issues or charitable causes, such as starvation, these might be perceived as less distant if, for example, people believe the consequences are more certain or already occurring. Research based on system justification theory suggests people are motivated to endorse the status quo, creating barriers to behaviour change or pro-environmental action[20]. Previous work on behavioural costs as a barrier to pro-environmental behaviour suggests these factors may interact[21–25]. Drawing on theoretical frameworks or empirical evidence of potential barriers to climate action is a common theme of several psychological interventions. Evidence suggests interventions that challenge these barriers or appeal to these drivers may positively influence pro-environmental attitudes and intentions[3]. However, for psychological interventions to impact real climate behaviours, it is vital to test whether they effectively change fully incentivized decisions, with a tightly matched control condition to establish specificity.

More broadly, controlled experimental designs and computational modelling techniques are vital to understand willingness to engage in effortful pro-environmental behaviors[26]. Some existing measures of pro-environmental behaviour, such as the Work for Environmental Protection Task[27] (WEPT) and the Carbon Emission Task[28] (CET), have started to include aspects of experimental design, including incentivising choices and using multiple trials. However, these tasks crucially lack a control condition. Previous work demonstrates the multidimensional nature of motivation—people can be apathetic to exert effort into some actions but not others. For example, they may be differentially willing to exert the same level of effort for the same amount of reward depending on who receives the reward[15–17]. Effective paradigms, therefore, need a control condition to test whether people are broadly motivated or motivated only for specific recipients or causes. The need for a control condition is further emphasised when testing intervention effects. The fact that existing paradigms lack a control condition makes it impossible to test whether interventions increase motivation to help all causes or specifically the pro-environmental motivation they were designed to affect. This distinction is key for understanding the mechanisms of successful interventions and potential implications of applied use of interventions.

Extensive work on motivation in behavioural science and neuroscience highlights multiple additional important design aspects that tasks should incorporate[9,10,14,29–31]. First, to understand the role of effort, it is crucial to control for time. Decisions should be between options that take the same amount of time and differ only in the effort required, to ensure that choices are not made based on the temporal discounting of rewards[32]. Second, the amount of effort required must be tailored to individuals' capacity for the effortful task, so differences between people are driven by motivation, not skill. Third, the reward magnitude and the amount of effort required should vary independently to quantify the effects of each, as for some people, incentives may matter more than effort or vice versa. Importantly, these strengths are not captured in existing measures such as the WEPT, which conflates cognitive and physical effort, time, and reward. While the CET does vary pro-environmental benefits and financial cost to oneself independently, the effort costs of benefitting the climate are not manipulated. It is impossible to exclude the possibility that more pro-environmental behaviour is actually due to a lower subjective valuation of financial costs.

Here, we present an effort-based decision-making task, the Pro-Environmental Effort Task (PEET), that integrates all these strengths to precisely capture willingness to exert effort for pro-environmental benefits, compared to identical choices to help a non-climate-relevant control cause. On each trial, participants chose between a no-effort, low-reward "rest" option and a high-effort, high-reward "work" option. On half of the trials, exerting effort helped the environment as rewards were donated to a charity that "prevents climate change by reducing carbon emissions". On the other half, rewards were donations to a control charity that "prevents starvation by providing food". If the participant chose the high-reward, high-effort work offer, they had to exert the required effort by clicking a button a set number of times within 10 s[33]. First, we aimed to validate the PEET in a large international sample by testing whether motivation to help the environment varies with reward and effort, as has been shown in other domains. Establishing this for climate motivation in a controlled experimental task also presents an important step in the literature. Our second aim was to use this design to test the effect of 11 interventions on climate motivation.

Importantly, the PEET paradigm allowed us to separately assess the impact of the reward available and the effort required, which were manipulated independently, on choices to help each cause. We hypothesised participants would help more when the reward was greater and when the effort required was lower. The design also enables us to fit computational effort-discounting models to the choice data to reveal the mechanisms behind decisions to help the climate. These quantify how the subjective value of the choice options (exert effort to help or rest) integrates the reward and effort, depending on how each participant discounts (or 'devalues') rewards by effort. The resulting participant-specific discounting $K$ parameters capture each participant's motivation. Crucially, computational models separate the influence of motivation (inverse of discounting) from consistency or decision noise, which can also affect choices. This allows us to test whether any successful climate interventions are increasing motivation, rather than simply changing how consistently people make choices. Using this paradigm, we conducted a pre-registered study (https://doi.org/10.17605/osf.io/zv2tu) with a large, international group of participants who completed the PEET (total $n = 3055$; samples from six countries recruited with the aim of being representative on age and gender).

Participants were randomly assigned to one of 12 groups, either a control group or one of the 11 pro-environmental psychological interventions developed by experts from the International Climate Psychology Collaboration (ICPC)[3,4] based on empirical and theoretical work. Each intervention used some or all of images, text, and asked participants to enter text (see Methods), based on the theme and previous research supporting it. We tested the effect of each intervention on two measures: (i) choices to exert effort, quantified as the percentage of times participants chose to exert effort rather than rest, and (ii) motivation, operationalised here as the inverse of the discounting $K$ parameters from the computational model. We hypothesised each intervention could increase choices to exert effort and motivation to benefit the climate compared to the food charity, relative to the control group who read a brief narrative unrelated to climate change. We also predicted individual variability in climate motivation (inverse $K_{climate}$) would be associated with climate-specific attitudes and beliefs as well as general measures of apathy and subjective effort.

## Methods

The study was preregistered on 21st November 2022: https://aspredicted.org/9fy2-3fyd.pdf.

## Participants

We recruited six samples through the Marketing Science Institute recruitment company as part of the International Climate Psychology Collaboration (ICPC)[3,4]. Country selection was based on increasing diversity relative to much research[34] and the research experience and linguistic expertise of the research team. Samples were recruited to be representative of gender and age distributions in each country (Bulgaria, Greece, Nigeria, Sweden, UK, USA). Participants who failed an attention check at the very start of the survey were immediately excluded and replaced with another participant. The following numbers of unique participants completed the first, ICPC part of the study: Bulgaria: 792, Greece: 827, Nigeria: 1528, Sweden: 2502, UK: 964, USA: 880. Of these, participants who failed a second attention check later in the study or did not correctly complete the WEPT demo were excluded by the ICPC based on preregistered criteria[3] (Bulgaria: 20, Greece: 149, Nigeria: 53, Sweden: 147, UK: 23, USA: 58). Unfortunately, a technical issue with the Greek version of the survey meant 532 participants

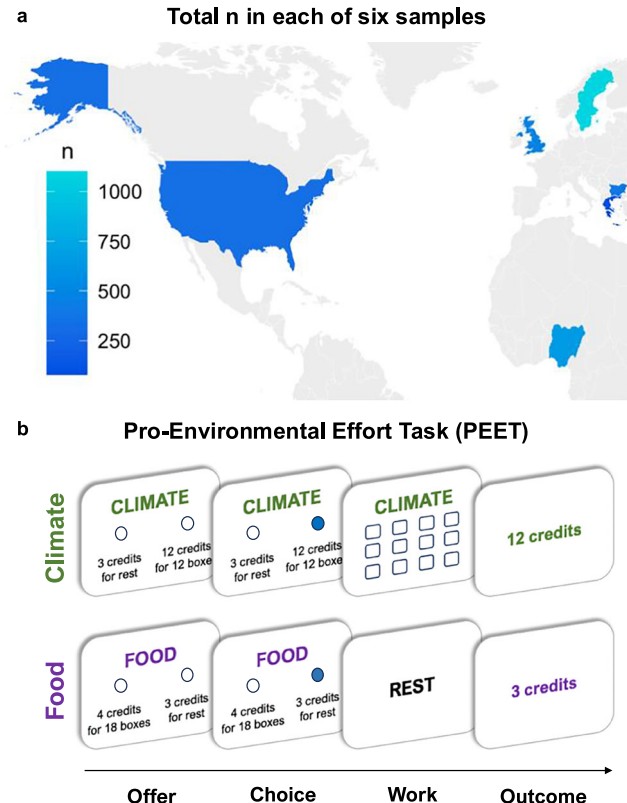

**a** Total n in each of six samples

n
1000
750
500
250

**b** Pro-Environmental Effort Task (PEET)

Climate

CLIMATE
3 credits for rest — 12 credits for 12 boxes

CLIMATE
3 credits for rest — 12 credits for 12 box

CLIMATE

12 credits

Food

FOOD
4 credits for 18 boxes — 3 credits for rest

FOOD
4 credits for 18 boxes — 3 credits for rest

REST

3 credits

Offer — Choice — Work — Outcome

**Fig. 1 | International sample and the Pro-Environmental Effort Task (PEET).**
**a** Participants from six countries across three continents completed the ICPC survey and the PEET: Bulgaria, Greece, Nigeria, Sweden, UK, USA. After applying preregistered exclusion criteria, across the six samples a total of 3055 participants were included in the analysis. **b** In the PEET, participants decide whether to exert effort for varying amounts of reward in the form of credits. Importantly, the credits obtained were real donations for two different charities—in half of the trials, the charity was climate-related (climate trials, top panel), and in the other half, it was non-climate related (food trials, bottom panel). Each trial started with a screen indicating the condition and the options—rest (no effort) for 3 credits and a work offer, associated with higher reward (4, 12, or 20 credits) for higher effort (50, 65, 80, or 95% of the boxes clicked in a calibration phase). Participants had 4 s to make a choice. If the work offer is chosen (top panel), participants need to click the specific number of boxes required to obtain the credits on offer in 10 s. If the rest option is chosen (bottom panel), participants rest for 10 s. Finally, the number of credits earned is displayed for one second, with zero credits earned following work choices but unsuccessfully meeting the required effort and following missed trials.

were excluded as information about the nature of the study was visible before the PEET. The number of eligible participants who reached the PEET experiment [against preregistered recruitment aims] was—Bulgaria: 727 [500], Greece: 146 [500], Nigeria: 1346 [1000], Sweden: 2056 [1500], UK: 856 [500], USA: 735 [500].

After receiving instructions about the PEET, participants completed two comprehension questions. If answered incorrectly, they saw reminders of the key aspects of the task and answered the questions again. As preregistered, we excluded participants who answered both questions incorrectly on the second attempt (Bulgaria: 82, Greece: 15, Nigeria: 191, Sweden: 305, UK: 145, USA: 149). We also excluded participants who missed more than 20% of trials in the PEET, in-line with our preregistration. While this resulted in a relatively large number of exclusions (Bulgaria: 241, Greece: 46, Nigeria: 495, Sweden: 661, UK: 229, USA: 252), it is important to ensure enough trials for analysis and because missing multiple trials could indicate a lack of engagement with the task. Therefore, the final analysis included a total of 3055 participants across six samples from Bulgaria: $n = 404$ (age 18–72, mean = 41.73, 195 female, 206 male, 3 other/unknown gender), Greece: $n = 85$ (age 19–61, mean = 37.05, 41 female, 43 male, 1 other/

unknown gender), Nigeria: $n = 660$ (age 18–68, mean = 32.27, 259 female, 401 male), Sweden: $n = 1090$ (age 18–74, mean = 42.84, 567 female, 512 male, 11 other/unknown gender), UK: $n = 482$ (age 18–74, mean = 47.96, 270 female, 211 male, 1 other/unknown gender), USA: $n = 334$ (age 19–74, mean = 47.78, 197 female, 135 male, 2 other/unknown gender; Fig. 1a). All participants provided informed consent and the study was approved by the following ethics review boards: University of Birmingham Science, Technology, Engineering and Mathematics (STEM) Ethics Committee (20-1897PA); The Ethics Committee of the Faculty of Business, Economics and Social Sciences of the University of Bern (232022); and University of Crete Research Ethics Committee (7875342DoPSS).

## Task and measures

**Pro-environmental effort task (PEET).** Participants decided whether to exert physical effort to earn money for a climate charity and a control food charity (Fig. 1B). Effort was quantified as clicking on-screen boxes. Before any instructions or information about the task, participants were prompted to click as many boxes as they could (of a maximum of 40) in 10 s. Participants then repeated this with encouragement to click even more boxes. The highest number across these two thresholding rounds was set as participants' maximum number of boxes used to threshold the effort levels throughout the experiment, with 13 boxes as the lowest maximum threshold. Next, participants read instructions about the PEET and completed five practice trials: four non-decision practice trials performing each effort level (i.e., 50, 65, 80, or 95% of their thresholded maximum number of boxes), and one decision trial identical to the ones in the main task. Finally, participants answered two comprehension questions about the task. If they answered either of these questions wrong, they received key information again and answered the same comprehension questions a second time.

On each trial, participants chose between a no-effort, low-reward (3 credits) "rest" option and a "work" offer with variable higher effort (50, 65, 80, or 95% of maximum effort) and higher reward (4, 12, or 20 credits). If they chose to work, the participant had to exert the required effort, i.e., clicking the indicated number of boxes within 10 s. If participants did so, they obtained the number of credits available. If they failed to do so, they did not get any credits for that trial. If participants chose the rest option, they rested for 10 s and obtained 3 credits. Participants had four seconds to select the work or rest option. If they did not, they had to wait 10 s with no credits obtained for that trial. The visual location of the work and the rest options was counterbalanced on the left or right side of the screen across trials.

Participants completed 24 trials in total, presented in a randomized order. For half of the trials, credits were for a climate charity, and the other half of trials could benefit a control, non-climate-relevant food charity. The descriptions of these charities were tightly matched, both endorsed by the United Nations, with the climate charity described as an organization that "prevents climate change by reducing carbon emissions," while the food charity "prevents starvation by providing food". Credits were converted into donations at the end of the study and made to the two charities.

**Work for environmental protection task (WEPT)**[27]. In the modified version of this task, participants made up to eight decisions of whether to screen a page of numerical stimuli for specific features (even first digit, odd second digit). Each completed page led to a tree being planted via donations to tree-planting organization. Participants were first exposed to a demonstration of the WEPT, identifying all target numbers with an even first digit and odd second digit. They then read information stating that planting trees is one of the best ways to combat climate change and that they would have the opportunity to plant up to eight trees if they chose to engage in additional pages of the task (one tree per completed page). Each page contained 60 numbers to screen for target numbers and displayed icons of eight trees, one of which was coloured green to mark their progress in the task. Participants were allowed to exit the task at any point.

**Climate beliefs[3]**. Participants rated four items in terms of "How accurate do you think these statements are?" (0 = not at all accurate to 100 = extremely accurate): "Taking action to fight climate change is necessary to avoid a global catastrophe", "Human activities are causing climate change", "Climate change poses a serious threat to humanity" and "Climate change is a global emergency". The measure had high internal consistency in the large ICPC sample[3,4] (Cronbach's alpha = 0.93, $n = 59,440$) and in the participants included in our analysis (Cronbach's alpha = 0.94, $n = 3055$).

**Climate policy support[3]**. Participants rated their level of agreement with nine statements (0 = not at all to 100 = very much so) on support for specific climate policies: "I support…" "…raising carbon taxes on gas/fossil fuels/coal", "significantly expanding infrastructure for public transportation", "increasing the number of charging stations for electric vehicles, "increasing the use of sustainable energy such as wind and solar energy", "increasing taxes on airline companies to offset carbon emissions", "protecting forested and land areas", "investing more in green jobs and businesses", "laws to keep waterways and oceans clean", and "increasing taxes on carbon intense foods (for example meat and dairy)". The internal consistency in the large ICPC sample[3,4] and the sample presented here was high (Cronbach's alpha = 0.88, n = 59,440; Cronbach's alpha = 0.89, n = 3055).

**Subjective effort ratings (NASA Task Load Index[35])**. Participants answered two questions asking how effortful they found the easiest and the hardest levels of effort using a 0–100 Likert scale.

**The Apathy Motivation Index (AMI)[36]**. Participants answered the 18 questions of the AMI, indicating their level of agreement with each statement. This scale comprises three subscales/domains of apathy: behavioural activation, emotional sensitivity, and social motivation.

### Interventions

**Working-together norms**. Participants read a flier promoting climate action as a collective effort, reinforcing the idea of working together with others to reduce carbon emissions.

**System justification**. Text and images framed climate change as a threat to participants' way of life and encouraged pro-environmental behaviour as patriotic.

**Binding moral foundations**. Participants read a message invoking national pride, loyalty, and authority to support clean energy and climate action.

**Exposure to effective collective action**. Participants were shown examples of successful climate-related movements to inspire hope and belief in the power of collective action.

**Future self-continuity**. Participants imagined a future version of themselves and wrote a letter to their present self about the importance of taking climate action now.

**Scientific consensus**. Participants saw a message and graphic emphasizing that 99% of climate scientists agree climate change is real and caused by humans.

**Decreasing psychological distance**. Climate change was presented as an immediate, local threat, and participants reflected on how it affects them personally.

**Dynamic social norms**. Participants read that more people are taking climate action globally over time, supported by examples and data showing behavioural trends.

**Correcting pluralistic ignorance**. Participants were shown how concern about climate change is much more widespread than people typically believe.

**Letter to future generations**. Participants wrote a letter to a future child or other family member, describing their efforts to protect the planet and how they wish to be remembered.

**Negative emotion**. Participants were exposed to emotionally intense, alarming climate information designed to induce negative emotions.

**Control group**. Participants read a neutral passage of text not related to climate change from *Great Expectations*.

### Procedure

All participants completed the experiment online via Qualtrics as part of the ICPC. For details of the ICPC collaboration procedure, intervention selection process, dataset, and results, see Vlasceanu, Doell, Bak-Coleman et al.[3] and Doell, Todorova, Vlasceanu et al.[4]. At the start of the study, participants saw a specific definition of climate change and were randomly assigned to one of 12 groups. In the control, no-intervention group, participants were exposed to non-climate content (passage of text from Great Expectations by Charles Dickens). In the other 11 groups, participants were exposed to an intervention crowd-sourced from academic experts (also see Supplementary Table 1 describing each intervention). Next, all participants answered a series of questions on their climate beliefs, climate policy support, willingness to share climate information (with the order of these three measures randomized between participants), then a modified version of the Work for Environmental Protection Task[27] (WEPT), and demographic information. The sample reported here from Bulgaria, Greece, Nigeria, Sweden, UK, and USA then completed the PEET, NASA ratings of subjective effort, and AMI (see above). The whole protocol, including the ICPC survey and the PEET with related measures, took approximately 30 min and was presented in the native language of each country, with English as an alternative language option.

### Statistics and reproducibility

We used R[37] (version 3.6.2) with R Studio[38] (version 1.4.1106) for analysis following our preregistered analysis plan. In line with our pre-registration, we analysed behavioural choice data and computational model parameters (see below and Supplementary Methods for full modelling information) with (generalized) linear mixed-effects models (LMM; *glmer*/*lmer* function; lme4 package[39] v1.1-27.1). Normality and equal variances were not formally tested as these models do not require data to strictly meet such assumptions, and the nature of the models account for the distribution of the data. Binomial GLMMs predicting people's decision to accept the high-effort high-reward work offer included within-subject fixed effects of reward available (level 2–4: 4, 12, 20 credits), effort required (level 2–5: 50, 65, 80, 95% maximum), and cause (climate vs. food). Random effects were grouped by participant nested in country and removed when necessary to obtain a converging model that maximizes power while minimizing Type I errors[40]. A fixed between-subject effect of intervention group and interaction between intervention and cause (climate / food) was then added to this model, making the final model:

$$choice \sim effort + reward + cause*intervention$$
$$+ (0 + effort + reward \,||\, country/participant)$$
$$+ (1 + agent \,||\, participant : country)$$

Models of computational discounting parameters ($K$) had fixed effects of charity (climate/food) and intervention (control group and 11 interventions), and a subject-level random intercept, as there is only one datapoint per participant per charity. The GLMMs of $K$s used a gamma distribution with log link function to account for the nature of the data without transforming raw values. Analysis of $\beta$ parameters used an LMM,

**Fig. 2 | Effort, reward, and cause determine choices to exert effort for the climate and food charities in the Pro-Environmental Effort Task (PEET). a** Participants in the control group, who did not experience a pro-environmental intervention, chose to exert physical effort for rewards more when the level of effort required was lower. **b** Choices to work were also higher when the reward available was larger. **c** Control participants were more willing to choose work to help the food charity compared to the climate charity. Asterisks between food and climate represent a significant effect of cause in the GLMM of choices (OR = 1.13 [1.02, 1.25], $p$ = 0.017). Shaded areas (**a**, **b**) represent 95% confidence intervals, error bars (**c**) show within-subject standard error, $n$ = 283 participants.

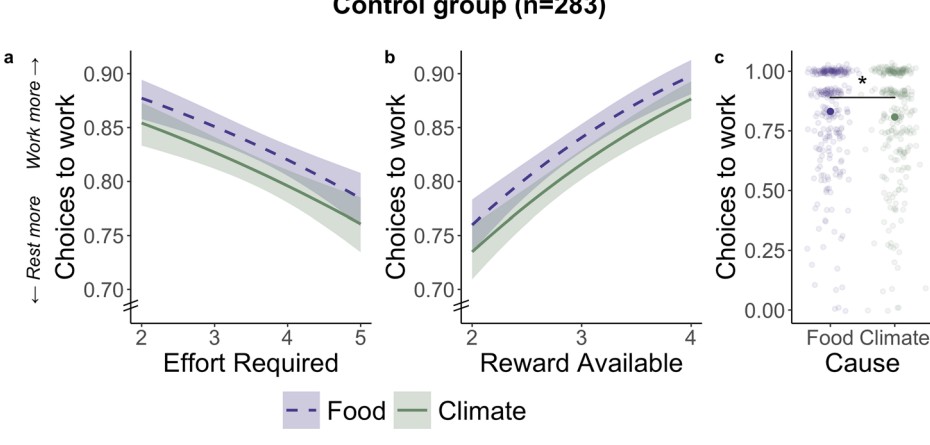

and choices to exert effort on the WEPT were analysed with cumulative link mixed models as previously[41], each with a fixed effect of intervention and subject-level random intercept. In all models, intervention was coded using treatment contrasts to compare each intervention to the control group reference, whereas cause was coded using sum-to-zero contrasts. Continuous variables were mean-centred. We applied a significance threshold of $p < 0.05$ for all fixed parameters in the models. We used the parameters package[42] (v0.18.1; *model_parameters* function) to extract standardized model coefficients (exponentiated in GLMMs to generate odds ratios for choices and mean ratios for $K$ parameters), their standard errors, and 95% confidence intervals. Bayes factors were calculated using the BayesFactor package (v 0.9.12-4.7; *ttestBF* function with default priors).

### Computational modelling

We quantified discounting of reward by effort ($K$) and decision consistency (inverse stochasticity $\beta$ parameter) by comparing multiple models that represent different plausible theories of discounting. All models had two, cause-specific parameters for discounting ($2K$: $K_{climate}$ and $K_{food}$) but varied in whether a single or two consistency $\beta$ parameters applied across causes ($1\beta$ or $2\beta$). We also varied whether the shape of the discount function was linear, hyperbolic, or parabolic[11], creating a total of six models (see Supplementary Methods). Models were fitted to the choice data using an iterative maximum a posteriori (MAP) approach as previously applied[43–45], implemented in MATLAB (2019b, The MathWorks Inc). See Supplementary Methods for full details of the MAP approach. All code for model fitting and simulations can be found at https://doi.org/10.17605/osf.io/zv2tu. Fitting the data across intervention groups using this method provides the most conservative comparison and is more robust to the influence of outliers than single-step maximum likelihood estimation[46]. It is therefore recommended over single step methods, where it is possible to implement[46].

### Model identifiability and parameter recovery

We used simulated data to establish that the model comparison procedure could correctly choose the best model and that parameters could be accurately estimated from our 24 trial schedule[47,48]. For model identifiability, we simulated data for 100 artificial agents based on each of the six models, drawing parameters randomly from a flat distribution between an upper and a lower bound covering all possible $K$ parameter values for that model ($0 < K < 1$ for linear, $0 < K < 2$ for parabolic) and $0 < \beta < 10$. Simulating ten datasets from each model and fitting each with the MAP approach and comparison procedure above generated confusion matrices showing the number of times the model was selected as best, based on exceedance probability. For parameter recovery, we simulated data using a grid of values covering the full ranges of the three parameters ($K_{climate}$, $K_{food}$, $\beta$) in the winning model across 176 simulated agents ($K$: 0, 0.3, 0.6, 0.9; $\beta$: integers 0–10) all with added noise drawn from a normal distribution * 0.05). As with model identifiability, we fit the simulated data using the MAP approach

applied to data from the participants and created a confusion matrix of the correlations between simulated and fitted parameter values.

### Reporting summary

Further information on research design is available in the Nature Portfolio Reporting Summary linked to this article.

## Results

We tested our preregistered hypotheses in samples from six countries, recruited to be representative on age and gender, creating a large international total $n$ = 3055 from Bulgaria ($n$ = 404, age 18–72, mean = 41.73, 195 female, 206 male, 3 other/unknown gender), Greece ($n$ = 85, age 19–61, mean = 37.05, 41 female, 43 male, 1 other/unknown gender), Nigeria ($n$ = 660, age 18–68, mean = 32.27, 259 female, 401 male), Sweden ($n$ = 1090, age 18–74, mean = 42.84, 567 female, 512 male, 11 other/unknown gender), UK ($n$ = 482, age 18–74, mean = 47.96, 270 female, 211 male, 1 other/unknown gender), USA ($n$ = 334, age 19–74, mean=47.78, 197 female, 135 male, 2 other/unknown gender; Fig. 1a). Participants were randomly assigned to one of 11 interventions or the control group[3,4] (see Methods, Supplementary Tables 1 and 2) and completed the PEET (Fig. 1b). First, participants clicked as many boxes as possible within 10 seconds to evaluate their maximum capability and all subsequent effort levels were tailored to this. Then on each trial, participants chose between a no-effort, low-reward "rest" option and a high-effort high-reward "work" option. On half of the trials, the reward was for an environmental charity that "prevents climate change by reducing carbon emissions". On the other half, they chose whether to exert effort to benefit a charity that "prevents starvation by providing food", providing a tightly matched non-climate control. The reward available (3 levels: 4, 12, 20 credits), effort required (4 levels: 50, 65, 80, or 95% of maximum), and cause (climate/food) were manipulated independently, allowing us to assess the impact of each and fit computational models that precisely quantify motivation to help each cause.

### Environmental benefits are devalued when they require effort

Our first analysis considered how the amount of reward available and the level of effort required affected decisions to take effortful actions that benefitted the environment or the food charity (hypotheses H1 and H2, analysis run as preregistered). We used generalized linear mixed-effects models (GLMMs) to determine whether choices between working and resting were sensitive to effort and reward, and additional follow-up tests to establish that these effects were found for both climate and food separately, as well as when combined. First, we included only participants in the control group, collapsed across all six countries, to quantify effort and reward effects in the absence of any intervention ($n$ = 283). As predicted, people were more willing to choose work when the effort required was lower (GLMM odds ratio (OR) [95% confidence interval] = 0.70 [0.58, 0.85], $p < 0.001$; Fig. 2a) and when the benefit was greater (OR = 1.96 [1.70, 2.27], $p < 0.001$; Fig. 2b

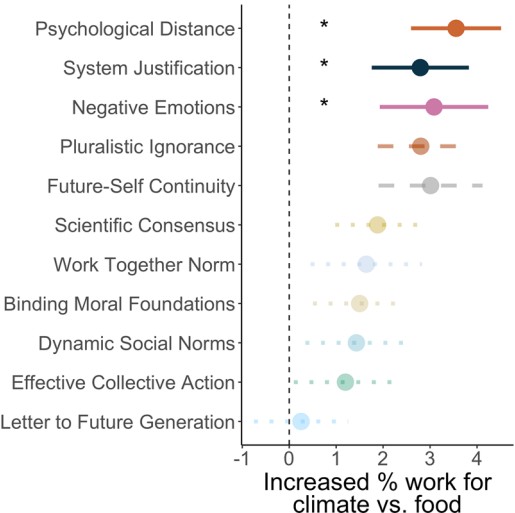

**Fig. 3 | Positive impact of interventions on choices to exert effort to help the climate relative to the food charity.** Comparing choices for each intervention against the control group showed consistent evidence that Psychological Distance, System Justification, and Negative Emotions interventions significantly increased relative willingness to work for the climate. There was also some evidence to support the effectiveness of the intervention based on Pluralistic Ignorance. Values plotted are the difference between each intervention and the control group (vertical line at zero) in choices to work for the climate – choices to work for the food charity. Significance was tested through cause (climate/food) by intervention (vs. control) interactions in GLMMs. Asterisks and solid lines show interventions with significant interaction effects that are consistent across all three control models (for age and gender, time on the survey before the PEET, and pages completed in the WEPT). Dashed lines show interventions with a significant effect in the main model or at least one control model. Dotted lines are interventions that do not significantly interact with the cause (climate/food) in any model. Error bars show standard error, n = 3055 participants.

and Supplementary Table 3). These significant effects of effort and reward were replicated in the full sample across intervention groups (effort OR = 0.76 [0.68, 0.84], $p < 0.001$; reward OR = 1.95 [1.66, 2.28], $p < 0.001$ and Supplementary Table 4) and in each country (Supplementary Fig. 1). In other words, people were more willing to choose effortful actions to protect the environment and provide food when the positive impact was greater, or the action was easier. These analyses also support that the PEET provides a rapid and robust tool to analyse sensitivity to effort and reward in choices to help the environment.

**Three psychological interventions increase relative willingness to help the environment**

After establishing that rewards for the climate and the food charity were devalued by the effort required to obtain them, we directly compared choices to work between the two tightly matched charities in the control group. Control participants showed a significant bias towards helping the food charity, over the climate charity (OR = 1.13 [1.02, 1.25], $p = 0.017$; Fig. 2c). This bias further emphasizes the importance of identifying interventions that boost actions to help the environment.

Next, we examined how the 11 interventions, developed based on psychological theories to facilitate pro-environmental actions and selected by experts, affected choices to put in effort to help the climate relative to the food charity (hypotheses H4). We tested the impact of each intervention compared to the control group using a GLMM (two-sided tests, alpha = 0.05 applied to each individual intervention effect; all aspects as preregistered see Methods). To establish results were robust, we also accounted for demographic characteristics that might affect choices to help, differing times since starting the study, and task engagement earlier in the study. Thus, we ran three control analyses: (i) age and gender, (ii) how much time had elapsed

since participants started the study, and (iii) pages completed on the WEPT (see Methods), an existing decision-making task that was completed before the PEET. For the latter, this existing decision-making task was always completed before the PEET, making it important to control for baseline differences in how much pro-environmental behaviour participants had already done in the study.

Of 11 interventions, three increased choices to help the climate charity relative to the food charity with effects consistent across control analyses: Psychological Distance (PD), System Justification (SJ) and Negative Emotions (NE; GLMM cause*intervention interactions: control vs. PD OR = 0.82 [0.71, 0.96], $p = 0.012$; control vs. SJ OR = 0.86 [0.74, 1.00], $p = 0.044$; control vs. NE OR = 0.85 [0.74, 0.99], $p = 0.036$; Fig. 3 and Supplementary Fig. 2). Participants who completed these interventions did not show the bias to help the food cause over the climate found in the control group, with Bayesian evidence of no difference between causes (Wilcoxon test PD V = 6004, $p = 0.17$, $BF_{01} = 6.76$; SJ V = 9222.00, $p = 0.75$, $BF_{01} = 13.76$; NE V = 6762.00, $p = 0.79$, $BF_{01} = 12.00$). A further intervention, Pluralistic Ignorance also significantly increased relative willingness to work for climate vs. food (OR = 0.86 [0.74, 1.00], $p = 0.0498$), but this was not consistent when controlling for age & gender (OR = 0.86 [0.74, 1.00], $p = 0.0503$) or time in the study (OR = 0.86 [0.74, 1.00], $p = 0.056$; Supplementary Table 5 for full results). Also see Supplementary Notes and Supplementary Table 6 for control analyses on WEPT performance in our sample to examine possible explanations for the discrepancy between our results and findings of Vlasceanu, Doell, Bak-Coleman et al.[3], where no intervention increased WEPT performance and some significantly decreased it.

Effect sizes for the interventions with significant effects showed approximately 3% increases in choices for climate relative to food (PD = 3.55%, SJ = 2.79%, NE = 3.08%; Fig. 3). To further strengthen the significance testing results and capture the precision of these effect size estimates we additionally conducted bootstrapping analysis with 1000 simulations of the dataset (see Supplementary Methods). Results showed that the medians of the simulation distributions closely captured our reported effects, providing confidence in their precision. Supporting the results of the interventions with significant effects in our main GLMM, the 95th percentiles of the simulated distributions did not include the value corresponding to no effect. In contrast, distributions (95th percentile) for the non-significant interventions included the null value (see Supplementary Notes, Supplementary Fig. 3, and Supplementary Table 7).

In summary, we identified several pro-environmental interventions that increased choices to exert effort to help the climate, relative to choices to help a non-climate cause. In particular, decreasing psychological distance between the participant and the changing environment, using system justification theory to promote pro-environmental action, and focusing on negative emotions about climate change increased relative willingness to help the environment.

**Computational models of effort discounting capture motivation to protect the environment**

Our model-free analysis showed choices to exert effort for the environment depended on the reward available and effort required. Next, we fitted computational models to the choices to capture the precise algorithm that integrates reward and effort into the subjective value of acting to help each cause. We fit models where the required effort devalued the available reward with different functions (linear, parabolic, hyperbolic), and choices depended on subject-specific parameters for effort discounting ($K$) and decision consistency ($\beta$ softmax function), as in previous work[9–11,14–17] and as preregistered. To enable testing our preregistered hypotheses of how discounting differs between causes, is affected by interventions, and relates to other measures, we deviated from the preregistered description of the analysis in only fitting models with two discounting parameters, one for rewards for the climate charity and the other for the food charity, rather than testing models with a combined discounting parameter. We compared models with a single $\beta$ parameter across causes to those with separate decision consistency parameters for climate and food (see Supplementary

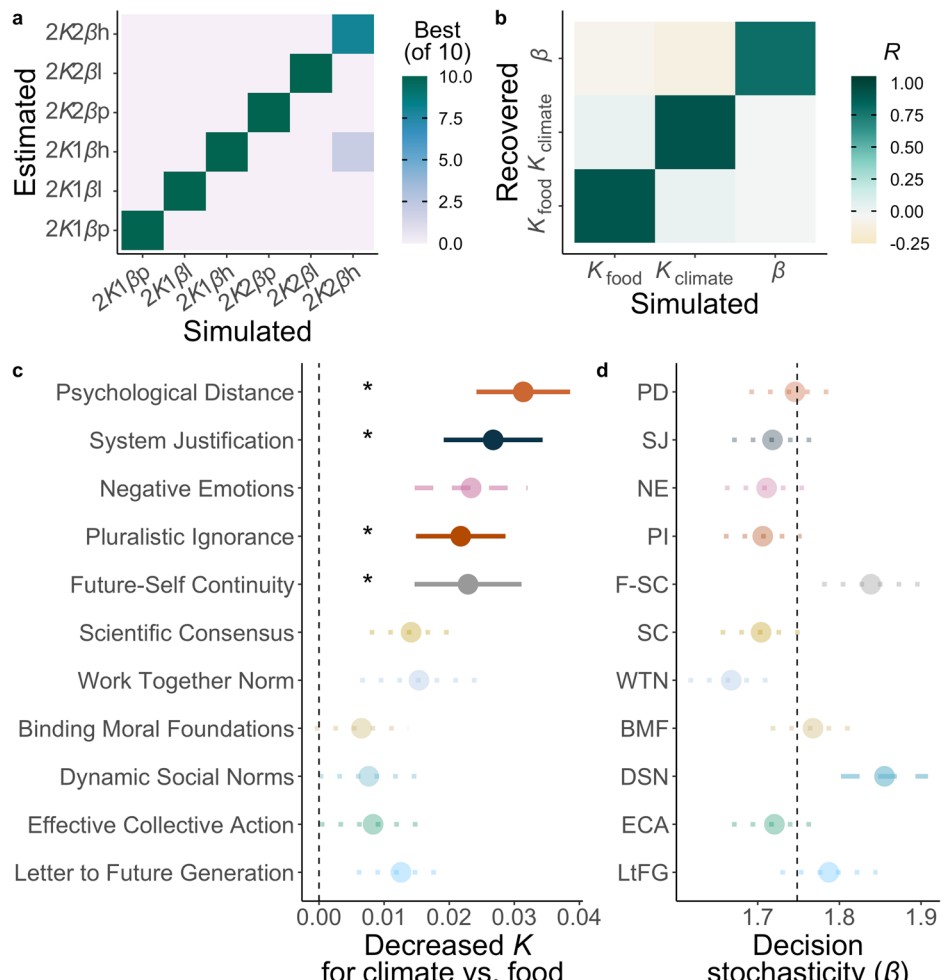

**Fig. 4 | Computational modelling reveals interventions increase relative motivation to help the climate.** We fit six computational models of effort discounting to the choice data. **a** Model identifiability analysis shows a strong diagonal across the winning model confusion matrix, meaning the models can be accurately identified and the model comparison process is robust. **b** Strong parameter recovery for the best fitting, $2K1\beta$-linear model is shown by the high correlations on the diagonal between simulated and recovered parameter values and low off-diagonal correlations. **c** Comparing discounting ($K$) parameters between causes for each intervention relative to the control group identifies four interventions consistently increase motivation (decrease discounting) for the climate vs. food: Psychological Distance, System Justification, Future-Self Continuity and Pluralistic Ignorance. Values plotted are the mean difference between each

intervention and the control group (vertical line at zero) in motivation for the climate–motivation for the food charity. **d** In contrast, comparing decision noise ($\beta$) for each intervention to the control group, only the Dynamic Social Norms intervention showed any evidence of an intervention effect, and this was not consistent across control analyses. Values are the mean $\beta$ parameter in each condition with the mean for the control group shown with a vertical line. Asterisks and solid lines show interventions with significant effects (GLMM cause*intervention for $K$, LMM intervention main effect for $\beta$) consistent across all three control models (age and gender, time on the survey before the PEET, and WEPT pages completed). Dashed lines are interventions with a significant effect in at least one model (main or control). Dotted lines are interventions with no significant effects. Error bars show standard error, $n = 3055$ participants.

Methods for full model space). We also established model identifiability across the six tested models (Fig. 4a) and showed strong parameter recovery for the winning model (Fig. 4b). Models were fit hierarchically (see Methods) to all the data, as this provides the most conservative test of any between-subject group differences.

The best-fitting model had a linear discounting function and a single $\beta$ parameter, as well as $K$ parameters for climate and food ($2K1\beta$-linear model: exceedance probability = 0.76, $R^2 = 0.76$, see Methods and Supplementary Table 8 for full results; see Supplementary Notes and Supplementary Table 9 for control analyses on parabolic discounting). The fact a single $\beta$ parameter best fit the data suggests that participants' choices followed subjective value in a similar way for both causes. The separate climate and food $K$ parameters, determined by the models tested, allowed us to assess differences in effort discounting between causes, how discounting was affected by interventions, and whether discounting of reward for the environment specifically relates to individual differences in pro-environmental attitudes.

## Changing pro-environmental motivation with psychological interventions

We extracted the discounting $K$ parameters from the best-fitting $2K1\beta$-linear model and compared motivation to help the climate with motivation to provide food, first in the control group. Participants had higher discounting parameters for the climate charity than the food charity (OR = 0.97 [0.94, 0.99], $p = 0.007$), demonstrating they devalued donations to the climate charity by the effort required to greater extent. We next examined if any intervention increased relative motivation to help the environment (hypothesis H4). Four interventions significantly reduced the food over climate bias (Fig. 4c) with consistent results across control analyses for age and gender, time in the study, and WEPT pages completed (Supplementary Table 10). Three of these interventions that decreased relative discounting also positively affected choices to work in the model-free analysis: Psychological Distance (OR = 1.06 [1.03, 1.10], $p < 0.001$), System Justification (OR = 1.05 [1.01, 1.08], $p = 0.008$), and Pluralistic Ignorance (OR = 1.04 [1.01, 1.08], $p = 0.022$). In addition, Future Self-Continuity also decreased

discounting (OR = 1.04 [1.00, 1.08], $p = 0.027$; see Supplementary Table 10 for full results). Again, these interventions completely eliminated the food over climate bias seen in the control group (Wilcoxon test PD $V = 15615$, $p = 0.40$, $BF_{01} = 8.76$; SJ $V = 19057$, $p = 0.95$, $BF_{01} = 13.17$; PI $V = 18543$, $p = 0.74$, $BF_{01} = 14.80$; FSC V = 10608, $p = 0.52$, $BF_{01} = 12.91$). Finally, as with the choices, bootstrapping analysis provided further evidence for these results and the precision of the effect sizes (see Supplementary Notes, Supplementary Fig. 4, and Supplementary Table 11).

Another strength of computational modelling is that $K$ parameters isolate discounting from decision consistency. We therefore evaluated whether interventions also changed decision consistency $\beta$ parameters. We found no evidence that such differences explained any of the significant intervention effects on choices or discounting (Fig. 4d and Supplementary Table 12). Only the Dynamic Social Norms intervention significantly differed from the control group, with greater consistency in choosing the option with higher subjective value following this intervention ($b = 0.13$ [0.02, 0.25], $p = 0.026$), but this was not consistent across control analyses (Supplementary Table 12).

In summary, computational modelling revealed select interventions significantly changed discounting of reward by effort in pro-environmental choices, independent of decision consistency. Interventions targeting Psychological Distance, System Justification, and Pluralistic Ignorance consistently decreased relative discounting of climate vs. food compared to the control group, just as they increased relative willingness to choose the work offer. There was also some evidence an intervention promoting a sense of continuity with one's future self effectively increased relative pro-environmental motivation.

### Motivation to take action for the climate is specifically associated with pro-environmental attitudes and beliefs

An additional advantage of computational modelling is that the separate discounting $K$ parameters quantify how each participant discounts rewards by effort. We term the inverse values of $K$ "motivation". This integrated measure can be used to examine how such motivation is related to individual difference in independent measures (hypothesis H3 and preregistered exploratory analysis of individual difference measures). In our final analyses, we combined data across interventions and tested whether motivation for effortful actions to help the environment (inverse $K_{climate}$) was associated with (i) climate-relevant measures of belief in human-made climate change and support for policies that protect the planet and (ii) general measures of trait apathy and subjective effort (see Methods). We also examined the same correlations for motivation to help the food charity (inverse $K_{food}$) to test the specificity of associations.

Motivation to exert effort for the climate (inverse $K_{climate}$) was positively associated with belief in climate change (negative correlation with $K$s $r_{(3055)} = -0.20$ [−0.23, −0.16], $p < 0.001$) and support for pro-environmental policies ($r_{(3037)} = -0.19$ [−0.22, −0.15], $p < 0.001$). Importantly, these associations between climate-relevant measures and pro-environmental motivation were significantly stronger than the corresponding correlations with motivation to help the food cause (belief $r_{(3055)} = -0.09$ [−0.12, −0.05], $p < 0.001$; difference $Z = 4.33$, $p < 0.001$; Fig. 5a; policy support $r_{(3037)} = -0.09$ [−0.12, −0.05], $p < 0.001$; difference $Z = 3.85$, $p < 0.001$; Fig. 5b). In contrast, broader measures of trait apathy and how subjectively effortful participants found the highest effort level, showed significant negative associations for both climate motivation and food motivation, with no significant difference between causes ($K_{climate}$ and apathy $r_{(2597)} = 0.10$ [0.06, 0.14], $p < 0.001$; $K_{food}$ and apathy $r_{(2597)} = 0.09$ [0.05, 0.13], $p < 0.001$; difference $Z = 0.31$, $p = 0.76$; Fig. 5c; $K_{climate}$ and subjective effort $r_{(2969)} = 0.13$ [0.09, 0.16], $p < 0.001$; $K_{food}$ and apathy $r_{(2969)} = 0.14$ [0.11, 0.18], $p < 0.001$; difference $Z = 0.55$, $p = 0.58$; Fig. 5d). Finally, we examined associations between pages completed in the WEPT and motivation to help each cause. WEPT performance was positively correlated with motivation for both the climate (negative correlation with $K$s $r_{(3055)} = -0.16$ [−0.20, −0.13], $p < 0.001$) and food ($r_{(3055)} = -0.18$ [−0.21,

−0.14], $p < 0.001$) charities, but importantly with no significant difference in the strength of these correlations ($Z = 0.54$, $p = 0.59$; Fig. 5e). Together these results highlight that general motivation is an important factor in increasing willingness to exert effort for any cause, but the PEET captures the specificity of variability in motivation to help the climate driving individual differences in climate-relevant outcomes. This specificity was not true for the WEPT.

## Discussion

Choosing actions that help the environment will be vital in limiting the devastating effects of climate change around the world. However, these actions are often more effortful than alternative behaviours, and as predicted by theories of effort aversion, people avoid effort. Here, we tested the effects of interventions designed to promote action to help the environment on the Pro-Environmental Effort Task in six samples from three continents. We found that the effort required and reward available influenced choices to work rather than rest, with people more likely to exert effort for a larger benefit, but less so when more effort was required. People also displayed an overall bias to exert effort for a food charity compared to a climate charity in the absence of intervention. However, several interventions, particularly those that targeted psychological distance and system justification, significantly reduced this bias, meaning people were relatively more willing to exert effort into actions that benefit the climate. Computational models of choices captured how reward and effort influenced motivation to benefit the climate or food cause, and separated this from how consistent people were in following these preferences. We showed that select interventions were effective on both people's choices as well as specifically on motivation to benefit the climate. In contrast, levels of decision consistency were similar between the food and climate conditions and could not explain the positive effect of interventions on choices. Finally, variability in motivation to help the environment was associated with individuals' belief in climate change and support for pro-environmental policies, whereas a lack of motivation across causes was associated with trait apathy and ratings of task effortfulness.

The finding that people were more willing to work for larger rewards, but less when the effort required was greater, builds on previous work on value-based decision-making in psychology, neuroscience and behavioural economics[9–16] as well as theories of effort aversion[13]. In the context of climate change, it fits with evidence of lower support for pro-environmental policies that involve personal costs[21–24] and behaviours perceived as more difficult[25]. Intriguingly, in real-world data, 7% of donations in the UK are to organizations working on conservation, the environment and heritage, compared to 11% for overseas aid and disaster relief[49]. This was mirrored in behaviour in our task as we found that participants in the control group were significantly more motivated to help a charity that provides food to prevent starvation than a charity that prevents climate change by reducing carbon emissions. Thus, effort and the aversion to exerting it are creating a barrier to engaging in pro-climate actions.

We found that two psychological interventions significantly increased relative willingness to help the climate across analyses and both model-free and model-based measures. One framed climate change as a proximal threat to decrease psychological distance and the other framed climate change as threatening the participants' way of life, based on system justification theory. The Psychological Distance intervention was based on findings that people perceive the consequences of climate change as uncertain, far in the future, affecting distant places, and impacting people different to themselves[18,19]. Previous work has suggested some mixed results on whether psychological distance is associated with pro-environmental attitudes and intentions, either through experimental manipulation or individual differences[50–52]. Psychological distance has also been suggested as a possible mechanism underlying the impact of experiencing extreme weather on climate behaviours[53]. In the International Climate Psychology Collaboration (ICPC) survey, the Psychological Distance intervention was the most effective at increasing belief in human-caused climate change, policy support, and willingness to share a pro-environmental message on social media[3]. Importantly, our results go further, showing an intervention that

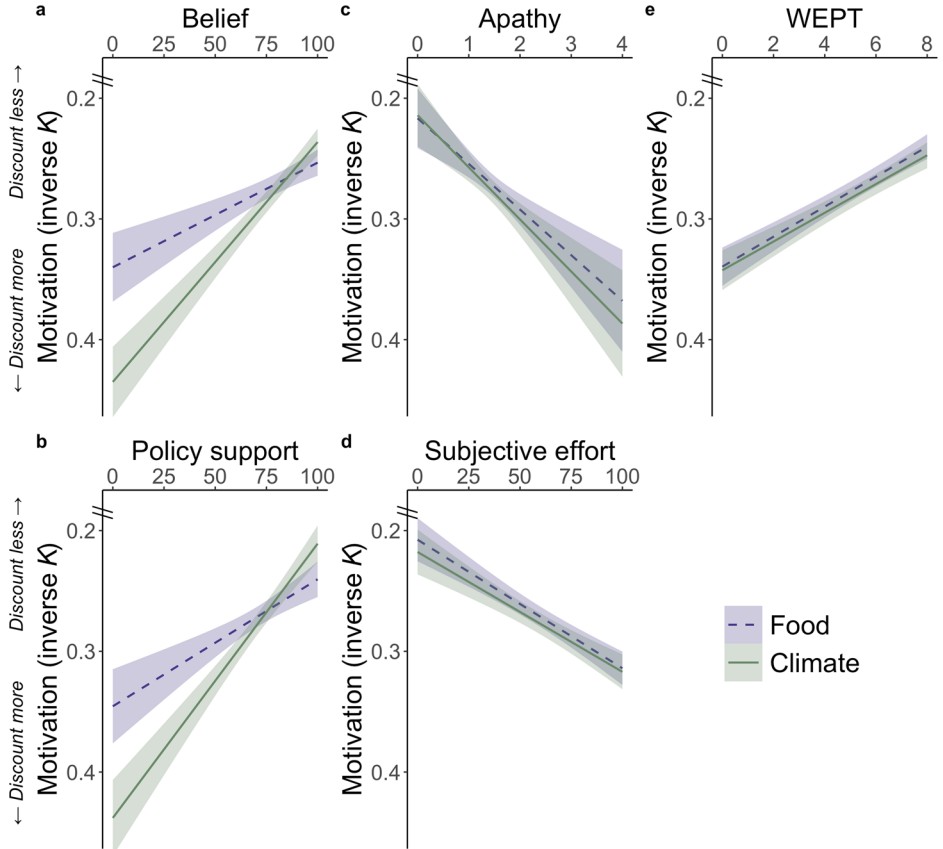

**Fig. 5 | Individual differences in pro-environmental motivation are associated with climate beliefs and policy support.** Motivation to help the climate, captured in lower discounting $K$ parameters, was associated with **a** beliefs that climate-change is human-made ($r_{(3055)} = -0.20$ [$-0.23$, $-0.16$], $p < 0.001$) and **b** support for policies that protect the environment ($r_{(3037)} = -0.19$ [$-0.22$, $-0.15$] $p < 0.001$). These positive associations between pro-environmental motivation ($K_{climate}$) and climate-relevant measures were significantly stronger than the corresponding association for motivation for the food charity ($K_{food}$; belief difference Z = 4.33, $p < 0.001$; policy support difference Z = 3.85, $p < 0.001$). Considering measures of general motivation, both **c** trait apathy and **d** self-reported ratings of subjective effort for the highest effort level were negatively associated with motivation for both the climate and food charity, with no significant

differences between these correlations (positive correlation between $K_{climate}$ and apathy $r_{(2597)} = 0.10$ [0.06, 0.14], $p < 0.001$; $K_{food}$ and apathy $r_{(2597)} = 0.09$ [0.05, 0.13], $p < 0.001$; difference Z = 0.31, $p = 0.76$; $K_{climate}$ and subjective effort $r_{(2969)} = 0.13$ [0.09, 0.16], $p < 0.001$; $K_{food}$ and apathy $r_{(2969)} = 0.14$ [0.11, 0.18], $p < 0.001$; difference Z = 0.55, $p = 0.58$). **e** the number of pages completed on the Work for Environmental Protection Task (WEPT) also positively correlated with motivation (lower $K$) to help both causes (climate ($r_{(3055)} = -0.16$ [$-0.20$, -0.13], $p < 0.001$; food $r_{(3055)} = -0.18$ [$-0.21$, $-0.14$], $p < 0.001$) with no significant difference between these correlations (Z = 0.54, $p = 0.59$). Shaded areas represent 95% confidence intervals, n = 3055 participants for (**a**, **e**), 3037 participants for (**b**), 2597 participants for (**c**) and 2969 for (**d**).

reduces psychological distance not only promotes pro-environmental attitudes, but also increases pro-environmental behaviour.

The second intervention that consistently increased pro-environmental motivation, System Justification, also aimed to eliminate a theorized psychological barrier to climate action. Previous work highlighted the importance of political orientation and ideology in predicting climate attitudes and beliefs[24,54] and these associations were explained by tendencies to justify and defend the current system[20]. The intervention uses the principles of system justification theory to frame climate change as threatening the participant's way of life, specific to their nation, and promote climate action as the patriotic response[20].

In addition to the consistent impact of Psychological Distance and System Justification interventions, there was some evidence supporting three further interventions: Negative Emotions, Pluralistic Ignorance, and Future Self-Continuity. The latter emphasized continuity across time through participants writing a letter from their imagined future self to their current self, describing pro-environmental actions to take now. It is interesting to note conceptual similarities in reducing perceived temporal and social distance between this intervention and the one directly targeting psychological distance. Future work could directly measure how distant participants perceive the consequences of climate change, compared to food

shortages, and consider the different components of distance, such as temporal, geographical, and social. The Pluralistic Ignorance intervention used real UN public opinion data to show participants most people in their country agree "climate change is a global emergency". This was based on findings that perceiving such beliefs as minority opinions reduced willingness to discuss climate change, but a similar intervention overcame this[55]. To the extent to which justifying the system relates to majority beliefs, this may have some similarities to the System Justification intervention. This mechanism could be followed up in future work.

Finally, the Negative Emotions intervention used "doom and gloom" framing of climate change impacts. This approach is controversial following mixed results on whether it increases or damages pro-environmental behaviour[56–59]. In the ICPC study, the Negative Emotions intervention was the most effective at increasing online sharing of pro-environmental content but significantly decreased willingness to exert effort in the WEPT[3]. Research to understand the complex associations between message framing, emotions, and attitudes or behaviours is ongoing in areas including charitable donations[60], portrayal of refugees[61], and consumption of online news[62]. Evidence that positive framing improved attitudes towards charitable appeals but negative messages increased actual donation behaviour[60] highlights the importance of measuring real incentivized choices. Further

work is needed to examine whether intervention effects are sustained over longer time periods[63] and how effectiveness differs between individuals and countries[3,41]. How to tailor interventions to individuals' baseline climate beliefs and motivation is an important open question[56].

A key strength of the current work and PEET is applying computational modelling to capture how rewards are discounted by effort in choices to help the environment. Comparing the effect of interventions between choices and discounting ($K$) parameters of motivation shows some minor inconsistencies, but overall, highly consistent results. While the measures are not identical and some variation in results is to be expected, the consistency suggests that differences in motivation are the mechanism by which interventions affect choices. We provide the most stringent test of interventions by focusing our conclusions on the two that show consistent effects across measures. By calculating discounting parameters for each individual, we revealed pro-environmental motivation is associated with both climate-relevant measures and broader trait motivation. While the broad trait measures were similarly associated with motivation to help the control food charity, there was specificity in how individuals' climate motivation linked to climate change belief and pro-environmental policy support. This suggests motivation to help the climate might be partially distinct from motivation to help other causes or individual people[15,16]. This distinction is important for predicting pro-environmental motivation from other variables. Interestingly, previous work found older adults were relatively more willing to exert effort to help another person[17], but also show stronger biases between charitable causes[64]. Existing evidence on the association between age and pro-environmental beliefs, attitudes, and intentions is mixed[24,65–68]. While our results were robust to controlling for age and gender, future work could examine differences based on these key demographic variables in both baseline climate motivation and how interventions impact willingness to take effortful actions.

Establishing the validity of the PEET also highlights advantages relative to existing measures. These additionally explain the discrepancy between our results (i.e., that several interventions increased effortful pro-environmental behaviour), and the negative or null effect of the same interventions on the WEPT shown in Vlasceanu, Doell, Bak-Coleman et al. (2024)[3]. First, our supplementary analyses revealed that intervention effects on PEET measures were robust to controlling for time, whereas effects on WEPT performance were not. In fact, the interventions that had negative effects on WEPT performance were those that took the most time[3]. The WEPT also showed a significantly higher ceiling effect (61% of participants) compared to the PEET (27% of participants). Checks for attention and understanding on the PEET further enhance robustness. We found evidence that the inclusion or exclusion of participants who failed these checks might account for the difference between previous findings of negative intervention effects[3] and the evidence here that several interventions increase climate motivation. Finally, while climate motivation measured by the PEET specifically linked to climate-relevant individual differences, the WEPT cannot show this specificity as it lacks a control condition. Performance on the WEPT was similarly associated with both climate and food motivation from the PEET, suggesting it may capture individuals' general tendency to complete tasks rather than being climate-specific. Together, these findings emphasise the need for a new tightly controlled experimental measure of pro-environmental motivation.

## Limitations

In addition to the strengths of the work, it is important to recognize limitations. We used a tightly controlled experimental task to precisely isolate and quantify the role of motivation to exert effort, but this may limit the external validity of the findings. Combining complementary approaches using experiments with a variety of real-world pro-environmental behaviours will be vital to fully understand how people choose to help mitigate climate change. This should include both actions that require more effort and high impact climate actions that do not require more effort than the alternative. Designs that measure motivation on the PEET pre- and post-intervention are needed to capture intervention effects within participant

and could then use overall climate motivation as the main outcome, rather than relative to the food charity. Future research should also test participants on the PEET without following other measures of pro-environmental attitudes and intentions. However, any demand characteristics that could be driven by the context of this study cannot explain results, as the control group was matched to the intervention groups on completing all of these measures. While results are based on six large samples from diverse countries, the countries were selected based on practical considerations. All were either European or English-speaking, and participants required a computer with internet. As preregistered, we accounted for country-level variance in our statistical analysis. However, we did not compare between countries to maintain power, and as there were no a-priori hypotheses on similarities or differences. Future research could examine climate motivation in more countries, and capture cultural differences with hypothesized links such as individualism-collectivism[41]. While challenging for experimental tasks, it would also be beneficial to recruit participants with less access to technology, often more likely to experience damaging consequences of climate change.

The two causes, food and climate, are not completely independent in everyday life, with natural disasters and food scarcity caused by climate change[1]. The interventions effects were also relatively small. Considering the raw difference in the percentage of times participants chose to work showed a 3% relative increase for the climate. Odds ratios from the models showed the effect of interventions on the odds of choices is 16–22% greater in the climate condition compared to the food condition for the successful interventions. For the odds of $K$s, this ranged from 4-6%. However, small changes to the everyday behaviours of millions of people make a sizable difference[69]. Understanding the mechanisms behind intervention effects may enable refining them for greater impacts[70]. Finally, we focus on individuals' choices, which are influential for carbon emissions, but addressing climate change also requires systemic changes in governments and businesses. Future work could examine increasing climate motivation in leaders and policymakers, and determinants of individuals' willingness to call for such change.

## Conclusions

In conclusion, using the PEET and computational approach, we identify how people make choices to exert effort to help prevent climate change. Willingness to work for environmental benefits increased when the effort required was lower or the rewards were greater. Computational modelling precisely captured how individuals devalued pro-environmental benefits by the effort required to obtain them. In the absence of intervention, participants were more willing to work to provide food to prevent starvation than reduce carbon emissions to mitigate climate change. However, several psychological interventions, particularly based on psychological distance and system justification, significantly reduced this bias and increased relative pro-environmental motivation. Finally, motivation to help the environment was associated with both climate-relevant and general traits. These results have important implications for understanding how people choose to act for the environment, how such motivation can be increased, and how it differs between people, critical advances in using behavioural science to limit the devastating impacts of the changing climate.

## Ethics and inclusion

The authors include representatives from five of the six countries included in the project, based on nationality and/or country of residence: Bulgaria— Boryana Todorova; Greece— Theofilos Gkinopoulos and Manos Tsakiris; Sweden—Lina Koppel and Daniel Västfjäll; UK—Jo Cutler, Luis Sebastian Contreras-Huerta, Katerina Michalaki, Matthew A. J. Apps and Patricia L. Lockwood; USA—Todd A. Vogel. The interventions, taken from the ICPC project, were tailored to each country, including translation, by the above representative(s) from each country. No co-authors or collaborators from Nigeria were involved in the project. Data from Nigerian participants were collected in English, meaning no translation was required, by Jo Cutler, Luis Sebastian Contreras-Huerta, Matthew A. J. Apps, and Patricia L. Lockwood in collaboration with the Marketing Science Institute, based on previous experience of data collection in Nigeria. Ethical approval for data collection

on the Pro-Environmental Effort Task (PEET) in each country was provided as follows: Bulgaria approved by the University of Bern (232022) as part of an ICPC data collection team covering German-speaking countries and Bulgaria. Greece approved by the University of Crete (7875342DoPSS). Nigeria, Sweden, UK and USA approved by the University of Birmingham (20-1897PA), covering data collection in the UK and abroad (would also cover Bulgaria and Greece). No additional efforts to obtain local approval were taken. The research does not result in stigmatization, incrimination, discrimination, or otherwise personal risk to participants. The research is globally relevant, including to each location included in the project.

## Data availability
Anonymised participant-level data, source data for all figures, and all materials required to run the Pro-Environmental Effort Task are available at: https://doi.org/10.17605/osf.io/zv2tu.

## Code availability
Code for modelling and analysis is available at: https://doi.org/10.17605/osf.io/zv2tu.

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

## Acknowledgements

This work was supported by a Wellcome Trust Early Career Award (227565/Z/23/Z) awarded to Jo Cutler; an ANID/FONDECYT Inicio grant (11250611) to Luis Sebastian Contreras-Huerta; funding (in part) from the Austrian Science Fund (FWF): ("DK Cognition and Communication 2" W1262-B29 [10.55776/W1262]; "Neuronal circuits in health and disease" COE16 [10.55776/COE16], and "The role of incentives and uncertainty in prosocial behavior" [10.55776/PAT1936023]) to Claus Lamm, funding from the Swedish Research Council and Knut and Alice Wallenberg Foundation to Daniel Västfjäll, the NOMIS Foundation grant for the Centre for the Politics of Feelings to Manos Tsakiris; a Christ Church Research Fund grant, a Biotechnology and Biological Sciences Research Council David Phillips Fellowship (BB/R010668/1), an ERC Consolidator (Je-S EP/Y014561/1) and a Wellcome Discovery Award (226645/Z/22/Z) awarded to Matthew A. J. Apps; and a Jacobs Foundation Research Fellowship, a Leverhulme Prize (PLP-2021-196), a Wellcome Trust/Royal Society Sir Henry Dale Fellowship (223264/Z/21/Z) and a UKRI EPSRC Frontiers Research Guarantee (EP/X020215/1) to Patricia L. Lockwood. The funders had no role in study design, data collection and analysis, decision to publish, or preparation of the manuscript. The authors would like to thank Dr Andrew Quinn for advice and guidance on the bootstrapping analysis.

## Author contributions

Conceptualisation: Jo Cutler, Luis Sebastian Contreras-Huerta, Matthew A. J. Apps, Patricia L. Lockwood Methodology: Jo Cutler, Luis Sebastian Contreras-Huerta, Boryana Todorova, Jonas Nitschke, Katerina Michalaki, Lina Koppel, Theofilos Gkinopoulos, Todd A. Vogel, Matthew A. J. Apps, Patricia L. Lockwood Investigation: Jo Cutler, Luis Sebastian Contreras-Huerta, Boryana Todorova, Jonas Nitschke, Katerina Michalaki, Lina Koppel, Theofilos Gkinopoulos, Todd A. Vogel Formal analysis: Jo Cutler, Luis Sebastian Contreras-Huerta Writing – Original Draft: Jo Cutler, Luis Sebastian Contreras-Huerta, Matthew A. J. Apps, Patricia L. Lockwood Writing – Review & Editing: All authors Funding Acquisition: Jo Cutler, Claus Lamm, Daniel Västfjäll, Manos Tsakiris, Matthew A. J. Apps, Patricia L. Lockwood Supervision: Claus Lamm, Daniel Västfjäll, Manos Tsakiris, Matthew A. J. Apps, Patricia L. Lockwood Please note authorship order is determined alphabetically for collaborators other than the joint-first and

joint-senior authors, grouped by early career researchers involved in methodology and investigation (Boryana Todorova, Jonas Nitschke, Katerina Michalaki, Lina Koppel, Theofilos Gkinopoulos, Todd A. Vogel) then principal investigators (Claus Lamm, Daniel Västfjäll, Manos Tsakiris).

## Competing interests

Patricia L. Lockwood is an Editorial Board Member for Communications Psychology, but was not involved in the editorial review of, nor the decision to publish this article. The other authors declare no competing interests.

## Additional information

[1]Centre for Human Brain Health, School of Psychology, University of Birmingham, Birmingham, UK. [2]Institute for Mental Health, School of Psychology, University of Birmingham, Birmingham, UK. [3]Centre for Developmental Science, School of Psychology, University of Birmingham, Birmingham, UK. [4]Center for Social and Cognitive Neuroscience, School of Psychology, Universidad Adolfo Ibáñez, Viña del Mar, Chile. [5]Center of Social Conflict and Cohesion Studies, Santiago, Chile. [6]Department of Cognition, Emotion, and Methods in Psychology, Faculty of Psychology, University of Vienna, Vienna, Austria. [7]Department of Psychology, Royal Holloway University of London, London, UK. [8]Centre for the Politics of Feelings, School of Advanced Study, University of London, London, UK. [9]Division of Economics, Department of Management and Engineering, Linköping University, Linköping, Sweden. [10]Institute of Psychology, Behavior in Crisis Lab, Jagiellonian University in Krakow, Krakow, Poland. [11]Division of Psychology, Department of Behavioral Sciences and Learning, Linköping University, Linköping, Sweden. [12]Birmingham Institute for Sustainability and Climate Action, University of Birmingham, Birmingham, UK. [13]Present address: Department of Social Sciences, School of Humanities and Social Sciences, University of Nicosia Athens Campus, Athens, Greece. [14]These authors contributed equally: Jo Cutler, Luis Sebastian Contreras-Huerta. [15]These authors jointly supervised this work: Matthew A. J. Apps, Patricia L. Lockwood. ✉e-mail: J.L.Cutler@bham.ac.uk; P.L.Lockwood@bham.ac.uk

