## [Transparent Peer Review file · Communications Psychology]

Psychological interventions that decrease psychological distance or challenge system justification increase motivation to exert effort to mitigate climate change

Corresponding Author: Dr Jo Cutler

Version 0:

Decision Letter:

Dear Jo,

Thank you for your patience during the peer-review process. Your manuscript titled "Psychological interventions increase motivation to exert effort to mitigate climate change" has now been seen by 2 reviewers, whose comments are appended below. I regret to inform you that in light of the referee reports, we cannot publish your manuscript in Communications Psychology.

You will see that the reviewers raise substantive concerns. Taking these points together with our editorial considerations, these reservations preclude publication of this study in Communications Psychology. Among other issues, both reviewers express novelty concerns. To be precise, the editorial concern in this regard is not that the analysis relies on published data - these are welcome - but that the analysis presented here does not present a sufficiently significant advance over the existing literature.

Although we cannot offer to publish your manuscript, I suggest that you consider Scientific Reports as a suitable venue for this work. To transfer your manuscript, please use our manuscript transfer portal. You will not have to re-supply manuscript metadata and files, unless you wish to make modifications. For more information, please see our [manuscript transfer FAQ](http://www.nature.com/authors/author_resources/transfer_manuscripts.html?WT.mc_id=EMI_NPG_1511_AUTHORTRANSF&WT.ec_id=AUTHOR) page.

I am sorry that we cannot be more positive on this occasion and thank you for the opportunity to consider your work.

Best wishes,
Marika

Marika Schiffer, PhD
Chief Editor
Communications Psychology

REVIEWERS' EXPERTISE:

Reviewer #1: environmental psychology, interventions

Reviewer #2: environmental psychology, interventions

REVIEWER COMMENTS:

Reviewer #1 (Remarks to the Author):

Review of Communications Psychology manuscript submission COMMSPSYCHOL-24-0641, "Psychological interventions increase motivation to exert effort to mitigate climate change".

The manuscript presents an original online study conducted in 6 diverse countries testing the impact of 11 crowd-sourced

interventions on the outcomes of a newly developed choice task that allows participants to invest real effort through mouse-clicking to earn rewards either for an environmental or a social charity. While I acknowledge the effort the authors have invested in conducting this study, I'm uncertain whether its findings significantly advance the field of environmental psychology. In particular, the theoretical contribution of the choice task, its validity, and advantage over existing measures remain unclear. Moreover, the design of the choice tasks does not seem to allow practical implications of the interventions beyond a context where resources are allocated between an environmental and a social cause. I detail my concerns below.

Major concerns

- 1) The reasoning for the inclusion of environmental, as well as food-security related effort tasks is unclear and potentially limiting the conclusions that can be drawn from the findings. Simultaneously completing both tasks may accentuate trade-off reasoning that might otherwise not influence individuals' choices (see work by Christopher Hsee on the influence of evaluation mode on preferences). Given that food crises are increasingly linked to natural disasters related to climate change, one could argue that the choice between the two reflects two frames of the same thing (with food potentially psychologically/socially closer than emission reductions).
- 2) The authors do not present any theoretical argumentation for the design of the novel choice task, how it might address shortcomings of other existing measures (carbon emission task, work for environmental protection task), nor do the authors present common validity checks, for example with other well validated measures of environmental behavior such as the GEB scale. The value of the novel choice task is therefore unclear and conclusions regarding the practical relevance of its outcomes appear overstated.
- 3) The authors argue that a focus on effort is needed, but do not discuss any related work looking into effort in the environmental domain or related concepts. There is, for example, a significant amount of literature on behavioral costs as a barrier to environmental behavior (e.g. work by Truelove and colleagues and also Kaiser). It would be important to understand where the authors place their research within this field of research and to extend the introduction and discussion accordingly.

Minor concerns

- 4) The authors employ relatively broad terms that mean something very specific in the field of motivation research but that are not shared by the wider community of researchers (even psychologists) studying environmental behavior. This leads to sub-headings that appear obvious, as for example in "Motivation to take action for the climate is specifically associated with pro-environmental attitudes and beliefs". Using more specific terminology instead of general terms like motivation would help the reader more clearly understand which outcomes are analysed in the different sections.
- 5) There should be a sharper distinction in the terminology used throughout the manuscript: devaluing and discounting environmental benefits; willingness to exert effort to help the climate and motivation to protect the environment.
- 6) More information on the content of the interventions is needed in the main manuscript.
- 7) Did the successful interventions completely eliminate the general preference for food over climate?
- 8) Although the authors did not hypothesise any differences between countries, a table showing the N's by condition and countries would be informative and useful.
- 9) The result presentation should more clearly distinguish between the choice outcomes: effort vs. rest (first part of analysis) and the subsequent variations in exerted effort at different levels of rewards/discounting. Explain in non-technical terms why this extra layer of analysis is interesting. Presently, the modelling findings do not present convincing evidence for why they are necessary to better understand environmental action.
- 10) How do the authors explain that findings are coherent for some interventions, but some interventions only seem to affect one of the outcomes?
- 11) Unclear why WEPT performance is included in the robustness check analyses. This doesn't seem to have been pre-registered and it's not clear to me at all how controlling for the performance in a similar consequential pro-environmental behavior task would make sense.
- 12) The authors could elaborate a bit more on what it means that the "differences in motivation were related to discounting reward by effort, not decision consistency". What would be the interpretation if decision consistency had explained invested effort?

Note on appeals: In exceptional circumstances, it is in authors' interest to appeal an editorial decision. More information on appeals is available here: <https://www.nature.com/commspsychol/submit/editorial-process#appeals>

Version 1:

Decision Letter:

Dear Jo,

Thank you for your correspondence asking us to reconsider our decision on your Article, "Psychological interventions increase motivation to exert effort to mitigate climate change". After careful consideration I have decided that we would be willing to consider a full appeal in the form of a revised version of your manuscript.

In your correspondence, you highlight that the reviewers may have misunderstood key aspects of the approach and therefore misjudged the methodological and conceptual advance that the work offers. Please be aware that we could only consider a revised version that alleviates all methodological concerns and is judged to offer significant insights on a conceptual level.

Along with your revised manuscript, you should also submit a separate point-by-point response to all of the concerns raised by the referees, in each case describing what changes have been made to the manuscript or, alternatively, if no action has been taken, providing a compelling argument for why that is the case.

Please note that we will only take the appeal forward and contact the reviewers again if we are persuaded that a substantial attempt has been made to address all editorial concerns and referees' comments. In this case, your revised manuscript and the point-by-point reply will be sent back to the referees so that they can judge whether their concerns have been addressed satisfactorily or otherwise.

I should stress, however, that we would be reluctant to trouble our referees again unless we thought that their comments had been addressed in full. If we do send the work out to review, we may seek additional feedback from a third referee.

When revising your paper please:

- ensure that it complies with the editorial policies outlined below
- ensure it meets our format requirements as set out in our [Guide to Authors](http://www.nature.com/commpsychol/submit/guide-to-authors) and highlighted below.
- ensure that the statistics reporting and interpretation is in line with journal guidelines <https://www.nature.com/commpsychol/submit/submission-guidelines#statistical-guidelines>

Please mark all correspondence via email with your Communications Psychology reference number in the subject line.

If the revision process takes significantly longer than five months, we will be happy to reconsider your paper at a later date, provided it still presents a significant contribution to the literature at that stage.

Please use the following link to submit your revised manuscript, point-by-point response to the Reviewers' comments with a list of your changes to the manuscript text (which should be in a separate document to any cover letter) and any completed checklist:

Link Redacted

Best wishes,

Marika

Marika Schiffer, PhD
Chief Editor
Communications Psychology

EDITORIAL POLICIES AND FORMATTING

Editorial Policy: [Policy requirements](https://www.nature.com/documents/nr-editorial-policy-checklist.pdf) (Download the link to your computer as a PDF.)

Furthermore, please align your manuscript with our format requirements, which are summarized on the following checklist: [Communications Psychology formatting checklist](https://www.nature.com/documents/commpsychol-style-formatting-checklist-article-rr.pdf)

and also in our style and formatting guide <https://www.nature.com/documents/commspsychol-style-formatting-guide-accept.pdf>>Communications Psychology formatting guide .

** Visit Nature Research's author and referees' website at <http://www.nature.com/authors>>www.nature.com/authors for information about policies, services and author benefits**

Version 2:

Decision Letter:

Dear Jo,

Thank you for your patience during the peer-review process. Your manuscript titled "Psychological interventions increase motivation to exert effort to mitigate climate change" has now been seen by the two previous reviewers, whose comments are appended below. We have discussed the reports and the details of your manuscript and decided that we can - conditionally - invite a revised version of the work.

You will see that the reviewers raise numerous concerns. Editorially, we consider the criticism regarding the corrections for multiple comparisons for individual interventions key. Reviewer #1 suggests the issue could be resolved presentationally. However, given the importance of substantive evidence as a basic criterion for publication in Communications Psychology, editorially this does not represent an appropriate solution (despite the other modelling results). We would not be able to take forward a manuscript that does not provide evidence for the intervention effects using a suitable correction method (which may for example be derived for example from model comparison). As pointed out by the Reviewer, the presentation leaves some uncertainty regarding the currently implemented approach (which should be clear from the description, without reference to the code).

To summarize, we would welcome a revision that addresses methodological concerns, but only provided an appropriate analysis supports the results. Alternatively, you may opt to transfer the work to our colleagues at Scientific Reports.

I have consulted with my colleague at Scientific Reports, and they have agreed to continue the review of your manuscript. We can issue you with a transfer link on request. For more information, please see our [a href=http://www.nature.com/authors/author_resources/transfer_manuscripts.html?WT.mc_id=EMI_NPG_1511_AUTHORTRANSF&WT.ec_id=AUTHOR/a](http://www.nature.com/authors/author_resources/transfer_manuscripts.html?WT.mc_id=EMI_NPG_1511_AUTHORTRANSF&WT.ec_id=AUTHOR)< page.

If you choose to transfer the manuscript to Scientific Reports, the manuscript should be revised to address all points raised by the referees. It would be handled by Scientific Reports in-house editors and sent for review to the original referees, if possible. The final decision would be guided by the advice received, and would not be based on the perceived impact of the study.

If you transfer the manuscript, please contact Scientific Reports Associate Editor, Dr Betty Mousikou (petroula.mousikou@springernature.com), with sre@nature.com in cc, as soon as you receive the new manuscript tracking number. Please also state in the cover letter that this is a transfer from Communications Psychology which was encouraged by Scientific Reports Associate Editor Dr Betty Mousikou.

I hope this provides the necessary clarity as you decide how to proceed.

Best regards,

Marika

Marika Schiffer, PhD
Chief Editor
Communications Psychology

REVIEWER COMMENTS:

Reviewer #1 (Remarks to the Author):

I want to thank the authors for their detailed explanations and the thorough revision of their manuscript. While the contribution of the present work is now clearer to me, I continue to have some important concerns, in particular related to the results presentation and interpretation.

Major concerns

1) In my view, the manuscript would substantially benefit from a clearer choice between focusing on presenting the properties of the novel task and the analysis of its outcomes. The section "Environmental benefits are devalued when they require effort" is interesting for a reader that is interested in the validity of the design of the task, but does not provide any novel insights. I suggest removing this section from the main manuscript and more clearly focus on the analysis of the task outcomes. Related parts in the discussion, that seem obvious to a large audience of scholars, could then also be removed (e.g., "people were more willing to work for larger rewards, but less when the effort required was greater").

2) I have concerns with regard to the results for the choice outcomes presented in the section "Psychological interventions increase relative willingness to help the environment" for which the hypothesized effects consistently fall right below the .05 alpha-level. In my view, this warrants extremely careful interpretation of these findings, and by no means claims about "robust evidence" (Figure 3).

a. Do these p-values correspond to one-sided tests of the interaction coefficients?

b. Would corrections for multiple comparisons be warranted given the considerable amount of interventions tested?

c. How does the selected random structure influence the results?

In my view, it would be appropriate to tone down any claims about the effects of the interventions on the choice outcome, and more strongly focus on the modeling results which appear more robust. The choice outcome results could for example be moved behind the modeling results. Unfortunately, I was not able to run the .Rproject shared through osf/git to verify some of the questions raised above.

3) The integration of the present work into the bigger survey on climate outcomes and including the WEPT is potentially problematic. It's a very particular setup to test a novel task after participants have spent about 20 minutes completing loads of other measures related to climate change and the environment. In particular, I wonder whether this might have emphasized consistent (pro-environmental) responding in the intervention groups and demand effects increasing the gap between relative preference for the climate over the food charity. This very particular circumstance should be more transparently discussed and its implications on the theoretical and practical implications that can be drawn from the collected data.

Minor concerns

4) Given that other parts might move to the supplementary material, I still think that more information on the content of the interventions is needed in the main manuscript.

5) The internal consistency within the used sample (and not the many labs) should be presented for the climate beliefs and climate policy support measures.

6) Use consistent terms for climate and food charity (not "control condition") from the start of the paper.

7) The terms "precisely" and "specifically" are used way too often throughout the manuscript and obstruct fluent reading without adding information or increasing the convincingness of the presented argumentation or evidence. I suggest to remove at least half of them.

8) Provide exact p value instead of $p < .05$ in footnote of Figure 2.

9) "Strikingly, control participants showed a significant bias towards helping the food charity, over the climate charity (OR=1.13 [1.02, 1.25], $p=0.017$;" I don't think that "striking" is the right word for this exploratory finding which just falls below the magic .05 alpha-level.

10) For all figures, display error bars for 95% confidence intervals instead of standard errors.

11) The authors should acknowledge that there are also high impact climate actions that do not require more effort.

Reviewer #2 (Remarks to the Author):

I thank the authors for their comprehensive responses to my queries about the added value, which I think are now adequately addressed. I still have key questions about the statistical analyses.

First, the authors did not correct for multiple testing. Correcting for multiple testing is appropriate if one (or more) significant effects out of the set is sufficient to reject the overall null hypothesis (e.g., Rubin, 2021). I think this applies here since the authors are rejecting their overall null hypothesis (i.e., interventions do not affect the relative effort exerted on the PEET) if just one of the interventions had a significant effect. In other words, the authors had 11 'opportunities' to reject the null hypothesis (see Rubin, 2021, p. 10978). Importantly, if the alpha level is adjusted to account for multiple testing, this renders all main findings in this paper non-significant ($0.05 / 11 =$ a significance threshold of 0.005). I am personally of the opinion that correcting for multiple testing would be appropriate here, also given the fact that the significant results reported by the authors float around 0.05, which makes me question their robustness. I would like to see a reflection from the authors on this.

It is also not clear to me why negative emotions and pluralistic ignorance 'flip places' between the first and second analysis. Specifically, negative emotions are reported as having a significant effect in the first analysis and not the second, and vice versa for pluralistic ignorance. Since these analyses are focussing on the same research question, I would expect the results to be similar across them.

Lastly, it is not clear to me why the authors focus on the exerted effort relative to the food condition as the main outcome of interest. I understand that this offers a relevant control, but the main effect on overall exerted effort in the climate condition seems like the primary outcome to me. The authors do analyze the overall exerted effort as an outcome but do not consider the effect of the interventions here. I do not follow the rationale here since the effect of the interventions is the main topic of this paper.

References:

Rubin, M. When to adjust alpha during multiple testing: a consideration of disjunction, conjunction, and individual testing. *Synthese* 199, 10969–11000 (2021). <https://doi-org.proxy-ub.rug.nl/10.1007/s11229-021-03276-4>

Version 3:

Decision Letter:

Dear Jo,

Your manuscript titled "Psychological interventions that decrease psychological distance or challenge system justification increase motivation to exert effort to mitigate climate change" has now been seen by our reviewers, whose comments appear below. In light of their advice I am delighted to say that we are happy, in principle, to publish a suitably revised version in *Communications Psychology*.

We therefore invite you to revise your paper one last time to address the remaining concerns of our reviewers and a list of editorial requests. At the same time we ask that you edit your manuscript to comply with our format requirements and to maximise the accessibility and therefore the impact of your work.

EDITORIAL REQUESTS:

SUBMISSION INFORMATION:

OPEN ACCESS:

*** TRANSPARENT PEER REVIEW:** *Communications Psychology* uses a transparent peer review system. On author request, confidential information and data can be removed from the published reviewer reports and rebuttal letters prior to publication. If you are concerned about the release of confidential data, please let us know specifically what information you would like to have removed. Please note that we cannot incorporate redactions for any other reasons.

*** CODE AVAILABILITY:** All *Communications Psychology* manuscripts must include a section titled "Code Availability" at the end of the methods section. We require that the custom analysis code supporting your conclusions is made available in a publicly accessible repository at this stage; please choose a repository that generates a digital object identifier (DOI) for the code; the link to the repository and the DOI must be included in the Code Availability statement. Publication as Supplementary Information will not suffice.

*** DATA AVAILABILITY:**

All *Communications Psychology* manuscripts must include a section titled "Data Availability" at the end of the Methods section. More information on this policy, is available in the Editorial Requests Table and at <http://www.nature.com/authors/policies/data/data-availability-statements-data-citations.pdf>.

Link Redacted

Best regards,

Marika

Marika Schiffer, PhD
Chief Editor
Communications Psychology

REVIEWERS' COMMENTS:

Reviewer #1 (Remarks to the Author):

I want to thank the authors for their thorough revision of the manuscript. In my view, their approach to responding to the multiple-testing concern raised by me and the other reviewer is unconventional, but convincing given their interpretation of only consistent results across both outcomes and all respective control analyses. All my concerns have been addressed.

Reviewer #2 (Remarks to the Author):

I appreciate the effort the authors have taken to respond to our comments. I think most of my comments have been addressed. I don't mean to delay the publication of the article, but the following minor points still came to mind.

First, I am still a bit puzzled by the inconsistent results across the section on the effect of the interventions on willingness to help the environment and the section on the K parameters. I would assume that decreasing K would be the mechanism that drives the increase in willingness to work for the environment, so it's not completely clear to me why these results do not align. Can you reflect on what this means for the validity of the task, or is it just a statistical artifact?

Second, I think a comparison to the results of Vlasceanu et al., (2024) would be appropriate. Specifically, they find that 'no intervention increased the more effortful behavior'. The current paper finds that two interventions 'had consistent positive effects on increasing effortful pro-environmental behaviours'. I think it would be worthwhile to discuss how the difference in tasks could explain why different results were obtained. The authors write that 'our results go further [than Vlasceanu et al., 2024], showing an intervention that reduces psychological distance not only promotes pro-environmental attitudes, but also increases pro-environmental behaviour'. I think this needs to be elaborated on more, especially since the original study also included a measure of more effortful pro-environmental behaviour.

Dear Marike,

Thank you for sending the comments from Reviewer #2 and for your time handling our manuscript, we realise you have many submissions to consider and appreciate you taking the time to send ours for review.

We are writing because we are concerned that the most substantive issues about the novelty of our contribution are based on a series of factual inaccuracies, and the comments suggest the reviewers have not closely read the paper. We also have further data that can directly address their comments and would like to enquire about the possibility of resubmitting a significantly updated manuscript.

Reviewer 2 states 'If I understand correctly, the data that are analyzed here are a subset of the full dataset examined in Vlasceanu et al., (2024). If this is the case, I am not convinced by the added value of the analyses reported in this paper.' This is in fact not the case and the reviewer has not understood correctly. The PEET task is completely novel and the data from it are not included in the work by Vlasceanu et al. (2024). We also have additional data that provides evidence against each of their possible criticisms of the task: (i) that participants do not perceive it as effortful (ii) that the task is too effortful so it becomes risky (iii) that it is not effortful enough to prevent a ceiling effect. We would be happy to provide these new data in a revised version of the manuscript.

Reviewer 1 suggests 'the theoretical contribution of the choice task, its validity, and advantage over existing measures remain unclear'. However, this information was indeed provided in the original manuscript. We very clearly establish the theoretical contribution (lines 16 to 33) and advantages of the task over existing measures (lines 35 to 49) in the introduction. We additionally provide empirical evidence of advantages, for example the existing WEPT measure lacks specificity that our novel PEET provides (Figure 5). It is also incorrect that we do not provide validity checks. We replicate the key effects measured by the PEET across countries (Figure S1) and show that climate motivation as captured by the PEET specifically links to measures of belief in climate change and policy support, which is not the case for the WEPT.

The fact that the reviewers have made these factually inaccurate statements about points that are actually very clear in manuscript makes us concerned about how closely they read it. Our significant advance over the existing literature is to present a validated novel task which, for the first time, allows computational modelling to precisely quantify motivation to exert effort, specifically for the climate compared to a closely matched control. Our task is also the first in the climate domain to threshold effort to individuals' own ability. We present evidence for intervention effects, not previously reported in Vlasceanu et al., 2024, and associations with trait measures. Therefore, our task and findings are indeed very novel and provide a substantial advance.

We would like to request if it is possible to appeal the decision on this manuscript and discuss a possible appeal over the phone or zoom?

Many thanks again for your time handling our manuscript.

In terms of additional information, we mentioned that we have data that provides evidence against each of Reviewer 2's possible criticisms of the task: (i) that participants do not perceive it as effortful (ii) that the task is too effortful so it becomes risky (iii) that it is not effortful enough to prevent a ceiling effect.

Briefly for each of the reviewer's points, our additional data show the following:

i) The reviewer states "I am thinking some participants may interpret this as a challenge or game and may therefore not perceive the task as effortful or depleting (especially in comparison to 'resting' which amounts to doing nothing)." In addition to the fact our task builds on thousands of previous studies showing that people avoid effort and find it aversive, we collected ratings of how effortful people found the lowest and highest effort levels, the NASA Task Load Index (Hart & Staveland, 1988). The significant associations between these ratings of subjective effort for the highest effort level and motivation for both food and climate (captured as discounting k parameters) is already presented in the manuscript. Examining the average values for these ratings on a 0-100 scale shows that participants rated the highest effort level 58.85 and the lowest effort level 37.36. These values also show that participants did perceive the highest effort level as significantly more effortful than the lowest with a large effect size ($d=0.81$ [0.77, 0.85], $p<0.001$). We also asked participants how tired they were before and after the PEET task. These data show the task was depleting, as participants got more tired between the start (mean=49.67) and end (mean=61.25) of the task ($d=0.48$ [0.44, 0.52], $p<0.001$).

ii) The reviewer states "The authors argue that fewer people chose to work on the task if the difficulty increased, which they argue shows that people are less likely to engage in pro-environmental behaviour if the effort increases. Yet, this effect may also be due to the fact that some participants may be considering the risk of not being able to complete the task in time and receiving zero credits against the certain reward of receiving 3 credits if they choose to rest." This explanation is very unlikely as the effort levels were thresholded to individuals' ability measured at the start of the task and the highest level was below 100% (95%) of what they performed during thresholding. Additional data also show that success rates were very high (climate mean=92.32%, food mean=91.54%) and comparable between the conditions. This means that risk is unlikely to account for behaviour in the task, but any small effect cannot explain differences between conditions or intervention effects.

iii) *The reviewer states “Figure 2 also shows participants chose to work on the task in most cases, even if the difficulty was very high. This also raises the concern of a ceiling effect.” In fact, only 38.25% of participants chose the work option on all of 24 PEET trials. For the WEPT task which both reviewers highlight as an established existing task, the proportion at ceiling (8/8 trials) is 61.05%. Furthermore, unlike the WEPT, all our analyses on the PEET data compare between conditions (climate vs. food). This allows us to reveal intervention effects specific to climate motivation, relative to motivation to help the control cause.*

Reviewer #1 (Remarks to the Author):

The manuscript presents an original online study conducted in 6 diverse countries testing the impact of 11 crowd-sourced interventions on the outcomes of a newly developed choice task that allows participants to invest real effort through mouse-clicking to earn rewards either for an environmental or a social charity. While I acknowledge the effort the authors have invested in conducting this study, I'm uncertain whether its findings significantly advance the field of environmental psychology. In particular, the theoretical contribution of the choice task, its validity, and advantage over existing measures remain unclear. Moreover, the design of the choice tasks does not seem to allow practical implications of the interventions beyond a context where resources are allocated between an environmental and a social cause. I detail my concerns below.

***Response:** Thank you for taking the time to review our manuscript. We are pleased you recognised our efforts in conducting a study with a novel and fully incentivised task in multiple countries. We appreciate your comments, which have helped us to improve our paper, clarify the task design, and highlight our contribution more clearly.*

Major concerns

1) The reasoning for the inclusion of environmental, as well as food-security related effort tasks is unclear and potentially limiting the conclusions that can be drawn from the findings. Simultaneously completing both tasks may accentuate trade-off reasoning that might otherwise not influence individuals' choices (see work by Christopher Hsee on the influence of evaluation mode on preferences). Given that food crises are increasingly linked to natural disasters related to climate change, one could argue that the choice between the two reflects two frames of the same thing (with food potentially psychologically/socially closer than emission reductions).

***Response:** Thank you for raising these points. First, we would like to clarify that the PEET is a single task with a climate condition and an independent food condition. Participants make choices between rest and exerting effort for the food cause, then on completely separate trials, choose between rest and exerting effort for the climate cause. There is never a 'choice between the two', and resources are never 'allocated between' the climate and food cause. Any potential trade-off, resting in one condition to save effort for the other condition, reflects the limits on effort and other resources that people have in the real world.*

Existing tasks, such as the Carbon Emission Task (CET) or the Work for Environmental Protection Task (WEPT), cannot show with specificity whether interventions increase climate motivation or simply change people's motivation across domains. By including a closely matched second cause (in our case a food charity) we can precisely test the specificity of the intervention effects, which reveals the underlying mechanisms that have changed, namely that people's relative willingness to benefit the climate cause has increased. In our individual differences analyses, we can then show that people's belief in climate change and policy support specifically correlate with motivation to exert effort to benefit the climate, more than motivation to benefit the food cause. We also can show how motivation for both climate and food correlate with number of pages completed in the WEPT. This reveals that the WEPT may index general motivation rather than specifically revealing people's preferences to benefit climate causes.

Rather than limiting conclusions, the additional condition allows us to make stronger conclusions in two ways. First, we show specificity of intervention effects on climate motivation rather than motivation for other charitable causes. Second, we underscore the need for a task that can test

and detect specificity through results that suggest the existing WEPT is not specific. We therefore demonstrate that our task and approach go significantly beyond existing approaches. The reasoning for including these two conditions was outlined in our original manuscript introduction (lines 45-59), but we have now further clarified and expanded this section:

“Previous work demonstrates the multidimensional nature of motivation – people can be apathetic to exert effort into some actions but not others. For example, they may be differentially willing to exert the same level of effort for the same amount of reward depending on who receives the reward¹⁵⁻¹⁷. Effective paradigms therefore need a control condition to test whether people are broadly motivated or motivated only for specific recipients or causes. The need for a control condition is further emphasised when testing intervention effects. The fact that existing paradigms lack a control condition makes it impossible to test whether interventions increase motivation to help all causes or specifically the pro-environmental motivation they were designed to affect. This distinction is key for understanding the mechanisms of successful interventions and potential implications of applied use of interventions.”

Second, we agree that food crises are linked to climate-related natural disasters and highlighted this in the original manuscript (lines 436-437):

“The two causes, food and climate, are also not completely independent in everyday life, with natural disasters and food scarcity caused by climate change¹.”

However, any overlap in how participants perceive the causes makes the specific intervention effects, and differential associations with individual differences, even more striking. Finally, the idea that food is potentially psychologically/socially closer than emission reductions is completely in line with our finding that an intervention that reduces perceived psychological distance robustly increases climate motivation. We have expanded on the relevant sections of the introduction and discussion to make this point clearer:

“Other suggested theoretical barriers to pro-environmental actions are that the consequences of climate change are psychologically distant: uncertain, far in the future, affecting distant places, and impacting people different to themselves^{18,19}. While some of these factors may also apply to other global issues or charitable causes such as starvation, these might be perceived as less distant if, for example, people believe the consequences are more certain or already occurring.”

“... and Future Self-Continuity. The latter emphasized continuity across time through participants writing a letter from their imagined future self to their current self, describing pro-environmental actions to take now. It is interesting to note conceptual similarities in reducing perceived temporal and social distance between this intervention and the one directly targeting psychological distance. Future work could directly measure how distant participants perceive the consequences of climate change, compared to food shortages, and consider the different components of distance such as temporal, geographical and social.”

2) The authors do not present any theoretical argumentation for the design of the novel choice task, how it might address shortcomings of other existing measures (carbon emission task, work for environmental protection ask), nor do the authors present common validity checks, for example with other well validated measures of environmental behavior such as the GEB scale. The value of the novel choice task is therefore unclear and conclusions regarding the practical relevance of its outcomes appear overstated.

Response: *We appreciate that there are multiple existing measures and agree it is important to justify the advantages and validity of any novel task. Indeed, we provided several sections on these points in the original manuscript. We very clearly established the theoretical contribution (lines 16-33) and advantages of the task over existing measures (lines 35-49) in the introduction. As outlined in these sections, we designed this task based on extensive theory on research on motivation and effort-based decision-making (e.g. Apps et al., 2015; Chong et al., 2017; David et al., 2022; Hartmann et al., 2013; Lockwood et al., 2017; Westbrook & Braver, 2015). In the revised manuscript we have expanded these sections to more explicitly contrast these strengths with shortcomings of existing measures (please see below).*

In the original manuscript, we provided empirical evidence of advantages over an existing measure, for example the WEPT lacks the specificity in associations with individual differences that our novel PEET provides (Figure 5). We also provided validity checks. We replicate the key effects measured by the PEET across countries (Figure S1) and show that climate motivation as captured by the PEET specifically links to self-reported measures of belief in climate change and policy support, which is not the case for the WEPT.

We have now also run additional analyses that further establish the validity and advantages of our task, as well as highlighting more shortcomings of the WEPT. We report the results of these analyses in a new section on task validity in the Supplementary Information.

The existing and revised sections of the manuscript that present the shortcomings of existing measures, advantages for the design of the PEET, and validity checks include:

Introduction

“One major challenge is that humans and other animals generally find effort aversive and avoid it, even when exerting effort obtains rewards, known as ‘the theory of least effort’ or effort aversion⁹⁻¹⁴. This aversion to effort is magnified in social contexts where the direct benefit is not immediately for ourselves, with people less willing to engage in effortful behaviours that help others¹⁵⁻¹⁷. Effort aversion may therefore be a significant barrier to actions that reduce climate change, which often do not have immediate benefits for the individual⁶. Other suggested theoretical barriers to pro-environmental actions are that the consequences of climate change are psychologically distant: uncertain, far in the future, affecting distant places, and impacting people different to themselves^{18,19}. Research based on system justification theory suggests people are motivated to endorse the status quo, creating barriers to behaviour change or pro-environmental action²⁰. Previous work on behavioural costs as a barrier to pro-environmental behaviour suggests these factors may interact²¹⁻²⁵. Drawing on theoretical frameworks or empirical evidence of potential barriers to climate action is a common theme of several psychological interventions. Evidence suggests interventions that challenge these barriers or appeal to these drivers may positively influence pro-environmental attitudes and intentions³. However, for psychological interventions to impact real climate behaviours, it is vital to test whether they effectively change fully incentivized decisions, with a tightly matched control condition to establish specificity.”

“More broadly, precisely controlled experimental designs are vital to understand the willingness to engage in pro-environmental behaviors²⁶. Some existing measures of pro-environmental behaviour, such as the Work for Environmental Protection Task²⁷ (WEPT) and the Carbon Emission Task²⁸ (CET), have started to include aspects of experimental design including incentivising choices and using multiple trials. However, these tasks crucially lack a control condition. Previous work demonstrates the multidimensional nature of motivation – people can be

apathetic to exert effort into some actions but not others. For example, they may be differentially willing to exert the same level of effort for the same amount of reward depending on who receives the reward^{15–17}. Effective paradigms therefore need a control cause, for example a non-environmental charity, to test whether people are broadly motivated or motivated only for specific recipients or causes. The need for a control condition is further emphasised when testing intervention effects. The fact that existing paradigms lack a control condition makes it impossible to test whether interventions increase motivation to help all causes or specifically the pro-environmental motivation they were designed to affect. This distinction is key for understanding the mechanisms of successful interventions and potential implications of applied use of interventions.

Extensive work on motivation in behavioural science and neuroscience highlights multiple additional important design aspects that tasks should incorporate^{9,10,14,22–24}. First, to understand the role of effort specifically, it is crucial to control for time. Decisions should be between options that take the same amount of time and differ only in the effort required, to ensure that choices are not made based on the temporal discounting of rewards²⁵. Second, the amount of effort required must be tailored to individuals' capacity for the effortful task, so differences between people are driven by motivation, not skill. Third, the reward magnitude and the amount of effort required should vary independently to quantify the effects of each, as for some people, incentives may matter more than effort or vice versa. Importantly, these strengths are not captured in existing measures such as the WEPT, which conflates cognitive and physical effort, time and reward. While the CET does vary pro-environmental benefits and financial cost to oneself independently, the effort costs of benefitting the climate are not manipulated. It is impossible to exclude the possibility that more pro-environmental behaviour is actually due to a lower subjective valuation of financial costs.”

Results

“Finally, we examined associations between pages completed in the WEPT and motivation to help each cause. WEPT performance was positively correlated with motivation for both the climate (negative correlation with Ks $r_{(3055)}=-0.16$ [-0.20, -0.13], $p<0.001$) and food ($r_{(3055)}=-0.18$ [-0.21, -0.14], $p<0.001$) charities, but importantly with no significant difference in the strength of these correlations ($Z=0.54$, $p=0.59$; Figure 5E). Together these results highlight that general motivation is an important factor in increasing willingness to exert effort for any cause, but the PEET captures the specificity of variability in motivation to help the climate driving individual differences in climate-relevant outcomes. This specificity was not true for the WEPT.”

Discussion

“Finally, establishing the validity of the PEET also highlights advantages relative to existing measures. While climate motivation measured by the PEET specifically linked to climate-relevant individual differences, the WEPT cannot show this specificity as it lacks a control condition. Performance on the WEPT was similarly associated with both climate and food motivation from the PEET, suggesting it may capture individuals' general tendency to complete tasks rather than being climate specific. Our supplementary results also revealed that intervention effects on WEPT performance were not robust to controlling for time, whereas effects on PEET measures were, and that the WEPT showed a significantly higher ceiling effect (61% of participants) compared to the PEET (27% of participants). These findings emphasise the need for a new tightly controlled experimental measure of pro-environmental motivation.”

Supplementary Results

“In the model of choices that controlled for number of pages completed on the WEPT, we additionally examined the association between this control variable and willingness to choose effort in the PEET. We found that participants who completed more pages on the WEPT were also more willing to exert effort in the PEET (OR=1.73 [1.59, 1.88], $p<0.001$). This positive association between PEET and WEPT measures was unexpected given we found that several interventions increased willingness to work in the PEET but previous results reported that these interventions did not increase motivation to complete the WEPT, in a sample that included our participants¹. We therefore examined intervention effects on pages completed in the WEPT using cumulative link mixed models (CLMM; see Methods) in all participants in the ICPC sample from the six countries (of 63 in total) we collected data in. As previously reported¹, no interventions significantly increased pages completed in the WEPT and several had significant negative impacts: Psychological Distance, Negative Emotions, Work-Together Norm, and Letter to Future Generation (Table S5). Therefore, differences between the two studies in how interventions increased pro-environmental effort were not due to our sample being unrepresentative or underpowered. Finally, we probed whether intervention effects would be observed when only including participants who completed the WEPT and also met pre-registered exclusion criteria for analysis of the PEET. Here, five interventions significantly *increased* the number of pages completed in the WEPT: Pluralistic Ignorance, Scientific Consensus, Collective Action, Dynamic Social Norms, and Binding Moral Foundations (Table S5). This result highlights the importance of adequate checks for attention and choice options with participants in the PEET having to decide between working or resting. We also note that controlling for time spent in the study did not impact on intervention effects in the PEET (Table S4), whereas controlling for the length of interventions also changed the effect of interventions on the WEPT results¹.

We additionally conducted several validity checks of the PEET to robustly establish the validity of our novel task. First, it could be argued that clicking boxes is not perceived as effortful or tiring but instead might be seen as a challenge or game. We therefore asked participants after the task to complete the NASA Task Load Index²² to test whether they perceived the highest effort level as significantly more effortful than the lowest and compared self-reports of tiredness after the PEET compared to before it (both ratings on 0-100 scales). Results showed that the highest effort level (mean rating=58.85) was perceived as significantly harder than the lowest effort level (mean=37.36, difference $d=0.81$ [0.77, 0.85], $p<0.001$). Similarly, participants were more tired at the end (mean rating=61.25) than before the start of the PEET (mean=49.67, difference $d=0.48$ [0.44, 0.52], $p<0.001$). Next, we considered the potential role of risk avoidance as an explanation for choosing rest over work. We compared the overall success rate after choosing to work in both conditions, as if success rates were low this would suggest risk was a factor driving participant choice. Instead, we observed high success rates in both conditions (climate mean=92.32%, food mean=91.54%) meaning when people did choose to work they were able to achieve the effort they chose. Finally, we examined the possibility of a ceiling effect in the PEET. This revealed 27.33% of participants chose the work option on all of 24 PEET trials. By comparison, for the WEPT task, the proportion at ceiling (8/8 pages completed) was significantly higher (61.05%, comparison $\chi^2_{(1)}$, $p<0.001$).”

3) The authors argue that a focus on effort is needed, but do not discuss any related work looking into effort in the environmental domain or related concepts. There is, for example, a significant

amount of literature on behavioral costs as a barrier to environmental behavior (e.g. work by Truelove and colleagues and also Kaiser). It would be important to understand where the authors place their research within this field of research and to extend the introduction and discussion accordingly.

Response: Thank you for drawing our attention to these additional papers. We did indeed highlight some related work on effort in the environmental domain in the original manuscript but have now expanded this including the work you cited:

Introduction

“Effort aversion may therefore be a significant barrier to actions that reduce climate change, which often do not have immediate benefits for the individual⁶. Other suggested theoretical barriers to pro-environmental actions are that the consequences of climate change are psychologically distant: uncertain, far in the future, affecting distant places, and impacting people different to themselves^{18,19}. Research based on system justification theory suggests people are motivated to endorse the status quo, creating barriers to behaviour change or pro-environmental action²⁰. Previous work on behavioural costs as a barrier to pro-environmental behaviour suggests these factors may interact²¹⁻²⁵. Drawing on theoretical frameworks or empirical evidence of potential barriers to climate action is a common theme of several psychological interventions. Evidence suggests interventions that challenge these barriers or appeal to these drivers may positively influence pro-environmental attitudes and intentions³. However, for psychological interventions to impact real climate behaviours, it is vital to test whether they effectively change fully incentivized decisions, with a tightly matched control condition to establish specificity.”

Discussion

“The finding that people were more willing to work for larger rewards, but less when the effort required was greater, builds on previous work on value-based decision-making in psychology, neuroscience and behavioural economics⁹⁻¹⁶ as well as theories of effort aversion¹³. In the context of climate change, it fits with evidence of lower support for pro-environmental policies that involve personal costs²¹⁻²⁴ and behaviours perceived as more difficult²⁵.”

Minor concerns

4) The authors employ relatively broad terms that mean something very specific in the field of motivation research but that are not shared by the wider community of researchers (even psychologists) studying environmental behavior. This leads to sub-headings that appear obvious, as for example in “Motivation to take action for the climate is specifically associated with pro-environmental attitudes and beliefs”. Using more specific terminology instead of general terms like motivation would help the reader more clearly understand which outcomes are analysed in the different sections.

Response: We apologise for any lack of clarity. In this work, we define and quantify motivation with the discounting K parameters from computational models. Lower values of K mean participants devalue rewards to a lesser extent by the effort required to obtain them and, therefore, have higher ‘motivation’. The discount rate is a common index of motivation in the decision-making literature (e.g. Chong et al., 2017; Hartmann et al., 2013; Lockwood et al., 2017). We have made our operationalisation of motivation clearer alongside improved definitions of the other terms

highlighted in point 5 – please see below. We have also edited the relevant figure legends to say “Motivation (inverse K)” rather than “Motivation (K)” and clarified the start of the section on associations of motivation with attitudes and beliefs:

“An additional advantage of computational modelling is that the separate K parameters precisely quantify how each participant discounts rewards by effort. We term the inverse values of K ‘motivation’. This integrated measure can be used to examine how such motivation is related to individual differences in independent measures. In our final analyses, we combined data across interventions and tested whether the motivation for effortful actions to help the environment (inverse K_{climate}) was associated with (i) climate-relevant measures of belief in human-made climate change and support for policies that protect the planet and (ii) general measures of trait apathy and current fatigue (see Methods). We also examined the same correlations for motivation to help the food charity (inverse K_{food}) to test the specificity of associations.

Motivation to exert effort for the climate (inverse K_{climate}) was...”

We would also point out that while the sub-heading highlighted may appear obvious, it is only with the novel PEET, which can test specificity of associations between motivation for a cause and individual differences, that such a conclusion can be experimentally supported.

5) There should be a sharper distinction in the terminology used throughout the manuscript: devaluing and discounting environmental benefits; willingness to exert effort to help the climate and motivation to protect the environment.

Response: *Thank you for the opportunity to clarify, we have now changed ‘willingness to exert effort’ to ‘choices to exert effort’ throughout the sections on choices so the terminology more closely links to the measure. We have also explicitly defined all terms for greater distinction and clarity (as well as making changes in response to point 9 below about the modelling):*

“With this design, we could separately assess the impact of the reward available and the effort required, which were manipulated independently, on choices to help each cause. It also enables us to fit computational effort-discounting models to the choice data to reveal the mechanisms behind decisions to help the climate. These precisely quantify how the subjective value of the choice options (exert effort to help or rest) integrates the reward and effort, depending on how each participant discounts (or ‘devalues’) rewards by effort. The resulting participant-specific discounting K parameters capture each participant’s motivation. Crucially, computational models separate the influence of motivation (inverse of discounting) from consistency or decision noise, which can also affect choices. This allows us to test whether any successful climate interventions are increasing motivation, rather than simply changing how consistently people make choices. Using this paradigm, we conducted a pre-registered study (<https://osf.io/zv2tu>) with a large, international group of participants who completed the PEET (total $n=3,055$; samples from six countries recruited with the aim of being representative on age and gender). Participants were randomly assigned to one of 12 groups, either a control group or one of the 11 pro-environmental psychological interventions developed by experts from the International Climate Psychology Collaboration (ICPC)^{3,4} based on empirical and theoretical work. We tested the effect of these interventions on two measures: (i) choices to exert effort, quantified as the percentage of times participants chose to exert effort rather than rest and (ii) motivation, operationalised here as the inverse of the discounting K parameters from the computational model. We hypothesized that at least some interventions would increase choices to exert effort and motivation to benefit the climate

compared to the food charity, relative to the control group who read a brief narrative unrelated to climate change. We also predicted individual variability in climate motivation (*inverse K_{climate}*) would be associated with climate-specific attitudes and beliefs as well as general measures of apathy and subjective effort.”

6) More information on the content of the interventions is needed in the main manuscript.

Response: *Information on the content of all 11 interventions is given in a full-page supplementary Table S1. Unfortunately, due to space restrictions, it is not possible to move this into the main manuscript. However, information about the content of the interventions of interest is provided in the existing manuscript.*

7) Did the successful interventions completely eliminate the general preference for food over climate?

Response: *Thank you for this interesting suggestion for an additional way to quantify the efficacy of the successful interventions. They did indeed eliminate the food over climate preference in all cases. We have added the results of these new post-hoc analyses to the manuscript:*

“Of 11 interventions, three increased choices to help the climate charity relative to the food charity with effects robust across control analyses: Psychological Distance (PD), System Justification (SJ) and Negative Emotions (NE; GLMM cause*intervention interactions: control vs. PD OR=0.82 [0.71, 0.96], $p=0.012$; control vs. SJ OR=0.86 [0.74, 1.00], $p=0.044$; control vs. NE OR=0.85 [0.74, 0.99], $p=0.036$; Figure 3 & Figure S2). *Participants who completed these interventions did not show the bias to help the food cause over the climate found in the control group, with Bayesian evidence of no difference between causes (Wilcoxon test PD $V=6004$, $p=0.17$, $BF_{01}=6.76$; SJ $V=9222.00$, $p=0.75$, $BF_{01}=13.76$; NE $V=6762.00$, $p=0.79$, $BF_{01}=12.00$).*”

Four interventions significantly reduced the food over climate bias (Figure 4C) with robust results across control analyses for age and gender, time in the study, and WEPT pages completed (Table S9). Three of these interventions that decreased relative discounting also positively affected choices to work in the model-free analysis: Psychological Distance (OR=1.06 [1.03, 1.10], $p<0.001$), System Justification (OR=1.05 [1.01, 1.08], $p=0.008$), and Pluralistic Ignorance (OR=1.04 [1.01, 1.08], $p=0.022$). In addition, Future Self-Continuity also decreased discounting (OR=1.04 [1.00, 1.08], $p=0.027$; see Table S9 for full results). *Again, these interventions completely eliminated the food over climate bias seen in the control group (Wilcoxon test PD $V=15615$, $p=0.40$, $BF_{01}=8.76$; SJ $V=19057$, $p=0.95$, $BF_{01}=13.17$; PI $V=18543$, $p=0.74$, $BF_{01}=14.80$; FSC $V=10608$, $p=0.52$, $BF_{01}=12.91$).*

8) Although the authors did not hypothesise any differences between countries, a table showing the N's by condition and countries would be informative and useful.

Response: *Thank you for this suggestion, we have added a table showing the sample size in each country and condition:*

Table S2. Number of participants in each intervention condition for each country

	Bulgaria	Greece	Nigeria	Sweden	UK	US	Total
Control	42	11	55	103	42	30	283
Work Together Norm	30	8	56	80	42	26	242
Negative Emotions	42	8	51	89	36	31	257
Scientific Consensus	33	11	58	101	43	25	271
Effective Collective Action	29	11	56	99	48	28	271
System Justification	31	7	62	92	43	40	275
Psychological Distance	32	0	56	93	41	20	242
Pluralistic Ignorance	36	9	60	98	38	34	275
Letter to Future Generation	30	5	47	76	28	17	203
Dynamic Social Norms	37	9	59	89	47	26	267
Future-Self Continuity	28	6	56	67	29	25	211
Binding Moral Foundations	34	0	44	103	45	32	258
Total	404	85	660	1090	482	334	3055

9) The result presentation should more clearly distinguish between the choice outcomes: effort vs. rest (first part of analysis) and the subsequent variations in exerted effort at different levels of rewards/discounting. Explain in non-technical terms why this extra layer of analysis is interesting. Presently, the modelling findings do not present convincing evidence for why they are necessary to better understand environmental action.

Response: Thank you for the opportunity to clarify. The second part of the analysis is not about the amount of effort exerted, it uses a computational model of the choices to exert effort vs. rest. For each participant and each choice, the model calculates the subjective value of each choice option (exert effort, rest), based on the reward available and effort required on that trial as well as the participant's level of discounting, quantified in the K parameter. The choice between options in the model then passes these two subjective values through a function that captures how consistently the participant selects their 'best' option (highest subjective value). Some participants might make choices that very consistently select their best option whereas others make noisier choices, sometimes choosing the worst option. By identifying the algorithm and participant-specific parameters that best explain choices, the computational modelling reveals the mechanisms behind pro-environmental choices.

We have clarified the details of the modelling in the revised manuscript (below). Changing 'willingness to exert effort to 'choices to exert effort' in response to point 5 above also helps the distinction. We have additionally more clearly outlined in the introduction why the computational modelling is necessary and interesting (and see below in response to point 10), to complement the sections of the existing manuscript that already mentioned strengths of the modelling approach:

"With this design, we could separately assess the impact of the reward available and the effort required, which were manipulated independently, on choices to help each cause. It also enables us to fit computational effort-discounting models to the choice data to reveal the mechanisms behind decisions to help the climate. These precisely quantify how the subjective value of the choice options (exert effort to help or rest) integrates the reward and effort, depending on how each participant discounts (or 'devalues') rewards by effort. The resulting participant-specific discounting K parameters capture each participant's motivation. Crucially, computational models

separate the influence of motivation (inverse of discounting) from consistency or decision noise, which can also affect choices. This allows us to test whether any successful climate interventions are increasing motivation, rather than simply changing how consistently people make choices.”

10) How do the authors explain that findings are coherent for some interventions, but some interventions only seem to affect one of the outcomes?

Response: Thank you for this interesting question. As outlined above, the modelling independently captures, and therefore separates, motivation (inverse discounting K parameter) and decision consistency β parameters. While it is fit to the choice data, we would expect some differences between motivation (inverse K) and the raw choices (percentage choose to exert effort). This is a key benefit of the computational modelling (also see above response to point 9) as it reveals which aspect of choices are driving any successful intervention effects.

Given the fact that the modelling removes variability in choices due to decision consistency, the intervention effects on choices and motivation (inverse K) are remarkably coherent. The interventions that only significantly increase relative climate motivation, across all control analyses, on one outcome do also affect the other outcome to some extent, the effect just does not quite reach statistical significance across all control analyses. For example, Negative Emotions has a robust significant effect on choices, and the effect on K parameters in the main analysis is $p=0.051$. Pluralistic Ignorance and Future-Self Continuity have significant effects on K parameters that are robust across control analyses, and for choices the p values across analyses were 0.048 to 0.056. No intervention significantly affected one outcome and did not have a significant effect on the other in any analysis. For example, Negative Emotions significantly affected K parameters when accounting for demographic variables ($p=0.044$). Coherence in the overall pattern of results is shown in Figure 4C. Here interventions are sorted by effect size on choices but the plot depicts the effects on K s, showing almost exactly the same order by size.

We have added this point to the revised discussion:

“However, several interventions, particularly those that targeted psychological distance and system justification, significantly reduced this bias, meaning people were relatively more willing to exert effort into actions that benefit the climate. Computational models of choices captured how reward and effort influenced motivation to benefit the climate or food cause, and separated this from how consistent people were in following these preferences. We showed that specific interventions were effective on both people’s choices as well as specifically on motivation to benefit the climate. In contrast, levels of decision consistency were similar between the food and climate conditions and could not explain the positive effect of interventions on choices. Finally, variability in motivation to help the environment was specifically associated with individuals’ belief in climate change and support for pro-environmental policies, whereas a lack of motivation across causes was associated with trait apathy and ratings of task effortfulness.”

11) Unclear why WEPT performance is included in the robustness check analyses. This doesn’t seem to have been pre-registered and it’s not clear to me at all how controlling for the performance in a similar consequential pro-environmental behavior task would make sense.

Response: *In order to test the association between the PEET and the WEPT at the individual level, all participants completed both measures with the WEPT always completed first. However, there is evidence that how much someone helps on one occasion can affect how likely they are to help at the next opportunity, for example the literature on moral self-licensing (e.g. Merritt et al., 2010; Blanken et al., 2015)). We therefore controlled for how much pro-environmental behaviour participants had already done before completing the PEET. Importantly, results were nearly identical between this control analysis, the other control analyses (for demographic factors or time since the start of the study) and the main analysis without these additional control variables. We have expanded our reasoning in the manuscript:*

“To establish results were robust, we also accounted for demographic characteristics that might affect choices to help, differing times since starting the study, and task engagement earlier in the study. Thus, we ran three control analyses: (i) age and gender, (ii) how much time had elapsed since participants started the study, and (iii) pages completed on the Work for Environmental Protection Task (WEPT; see Methods). For the latter, this existing decision-making task was always completed before the PEET, making it important to control for baseline differences in how much pro-environmental behaviour participants had already done in the study.”

12) The authors could elaborate a bit more on what it means that the “differences in motivation were related to discounting reward by effort, not decision consistency”. What would be the interpretation if decision consistency had explained invested effort?

Response: *Thank you for the opportunity to elaborate. The computational model captures two types of parameters: first how much people value rewards, discounted by how much effort they have to put in to get them (K parameters) and second, how consistent people are in following their value preferences. If we had found that the interventions affected the decision consistency parameter, this would mean the interventions changed how likely people were to follow their preferences, not how much they valued the benefits for the climate discounted by effort. This would lead to quite a different interpretation of interventions effects, with different implications for how the mechanisms of the intervention impacted on people’s decisions.*

“However, several interventions, particularly those that targeted psychological distance and system justification, significantly reduced this bias, meaning people were relatively more willing to exert effort into actions that benefit the climate. Computational models of choices captured how reward and effort influenced motivation to benefit the climate or food cause, and separated this from how consistent people were in following these preferences. We showed that specific interventions were effective on both people’s choices as well as specifically on motivation to benefit the climate. In contrast, levels of decision consistency were similar between the food and climate conditions and could not explain the positive effect of interventions on choices. Finally, variability in motivation to help the environment was specifically associated with individuals’ belief in climate change and support for pro-environmental policies, whereas a lack of motivation across causes was associated with trait apathy and ratings of task effortfulness.”

Reviewer #2 (Remarks to the Author):

This article reports data from 6 countries that examined whether different types of psychological intervention can increase the effort participants are willing to exert on a task in order to help an

environmental organization. The findings reported show that three out of 11 interventions increased effort expended to help a climate charity, namely Psychological Distance (PD), System Justification (SJ) and Negative Emotions.

Response: *Thank you for your time reviewing our work and your comments that have helped us improve the manuscript.*

I have a few questions about how the data in this paper relate to the results reported in Vlasceanu et al. Addressing climate change with behavioral science: A global intervention tournament in 63 countries. *Sci. Adv.* 10, ead5778 (2024). If I understand correctly, the data that are analyzed here are a subset of the full dataset examined in Vlasceanu et al., (2024).

If this is the case, I am not convinced by the added value of the analyses reported in this paper. The original analysis of Vlasceanu examines the same research question but with a much larger set of data across more countries. It is not clear what the added value of the results reported here is beyond the findings already reported in Vlasceanu et al., 2024.

Response: *Apologies for any confusion. It is not the case that the data analysed in the present manuscript are a subset of those reported in Vlasceanu et al., (2024). In the 'many labs' approach used by Vlasceanu et al., research teams in each country collected data on measures including the WEPT following the interventions and could add additional measures in for their samples. In the countries we sampled, additional measures were the PEET and multiple other questionnaires. Thus, our data is from some of the same participants as reported in Vlasceanu et al., but the PEET task is completely novel and the data from it have never been published. Instead, our results reveal completely different conclusions to Vlasceanu et al., (2024). We observed using our novel task that some interventions increase motivation to exert effort to benefit climate causes. In Vlasceanu et al., (2024), using the WEPT, they concluded climate interventions reduced pro-environmental behaviour.*

We have made this clearer in the revised manuscript, as outlined below.

I notice there are two main differences compared to Vlasceanu et al., (2024), but in my opinion these do not represent a sufficiently novel addition to the findings already reported. First, the authors added a control by also including trials focused on a non-climate related cause. I think this offers some additional evidence to Vlasceanu et al., (2024), but the contribution is modest.

Response: *Thank you for the opportunity to clarify. As detailed above, and as you state below, our paper uses the PEET, a novel task which is different from the WEPT. The key differences between these tasks include a non-climate control condition as well as important design features where, unlike the WEPT, the PEET:*

- *Measures willingness to exert physical effort specifically, rather than both cognitive and physical effort. This is important as people can differ in their willingness to exert cognitive compared to physical effort.*
- *Calibrates individuals' ability to exert effort and tailors the levels used in the task*
- *Does not conflate effort and time costs*
- *Independently manipulates the effort level and reward level*

- *Has repeated, within-subject trials creating a continuous measure of willingness to exert effort rather than a 1-8 scale*
- *Is robust to controlling for time*

We believe all of these are crucial aspects of experimental design and ways that the PEET improves on existing measures. Regarding a non-climate control condition specifically, this is vital to test whether the effects of interventions or associations with individual differences are specific to pro-environmental motivation or apply to motivation generally. Work testing whether healthy eating interventions increase how many vegetables people eat would need to show this was not simply making them eat more of all foods. While helping a non-climate charity is still a positive behaviour, testing whether pro-environment interventions specifically affect pro-environmental motivation is key to advancing understanding in this field. The food control condition also enables us to demonstrate specificity in associations with individual differences. Climate motivation was specifically linked with climate beliefs and policy support, whereas motivation to help both causes was equally linked to general apathy. These comparisons are not possible with the WEPT, which only has a climate condition. However, interestingly, WEPT scores were equally associated with motivation to help the food cause and climate cause, suggesting that the measure might in fact capture general motivation, rather than willingness to help the climate specifically.

While our original manuscript did outline the advantages of our task and the importance of a non-climate control condition we have edited, clarified and expanded on these points – please see below for all changes also in response to your following queries.

Second, the task to assess pro-environmental behaviour in this study (PEET) is different from Vlasceanu et al., (2024), which used the Work for Environmental Protection Task (WEPT). I am not (yet) convinced the PEET represents a significant advance over the WEPT. If I understand correctly, for the PEET task participants had to click on a certain amount of boxes within a time limit. I am thinking some participants may interpret this as a challenge or game and may therefore not perceive the task as effortful or depleting (especially in comparison to ‘resting’ which amounts to doing nothing). How is the ecological validity of this task established? The authors argue that fewer people chose to work on the task if the difficulty increased, which they argue shows that people are less likely to engage in pro-environmental behaviour if the effort increases. Yet, this effect may also be due to the fact that some participants may be considering the risk of not being able to complete the task in time and receiving zero credits against the certain reward of receiving 3 credits if they choose to rest. The validity of the task is therefore not fully established in my view. Figure 2 also shows participants chose to work on the task in most cases, even if the difficulty was very high. This also raises the concern of a ceiling effect.

Response: *Thank you for raising these important considerations about the PEET task, we address each in turn.*

First, there is a question of whether participants interpret clicking boxes as a challenge or game and may therefore not perceive the task as effortful or depleting. In addition to the fact our task builds on hundreds of previous studies, with thousands of participants across multiple species, showing that people and animals avoid effort and find it aversive (e.g. Apps et al., 2015; Chong et al., 2017; David et al., 2022; Hartmann et al., 2013; Lockwood et al., 2017; Westbrook & Braver, 2015), we collected ratings of how effortful people found the lowest and highest effort levels, the NASA Task Load Index (Hart & Staveland, 1988). The significant associations between these

ratings of subjective effort for the highest effort level and motivation for both food and climate (captured as discounting K parameters) is already presented in Figure 5d. Examining the average values for these ratings on a 0-100 scale shows that participants rated the highest effort level 58.85 and the lowest effort level 37.36. These values also show that participants did perceive the highest effort level as significantly more effortful than the lowest with a large effect size ($d=0.81$ [0.77, 0.85], $p<0.001$). We also asked participants how tired they were before and after the PEET task. These data show the task was depleting, as participants got more tired between the start (mean=49.67) and end (mean=61.25) of the task ($d=0.48$ [0.44, 0.52], $p<0.001$).

Second, the issue of whether choices are driven by a risk of not being able to complete the task in time and receiving zero credits against the certain reward of receiving 3 credits if they choose to rest. This explanation is very unlikely as the effort levels were thresholded to individuals' ability measured at the start of the task and the highest level was below 100% (95%) of what they performed during thresholding. Additional data also show that success rates were very high (climate mean=92.32%, food mean=91.54%) and comparable between the conditions. This means that risk is unlikely to account for behaviour in the task, but even if there was a small effect this cannot explain differences between conditions or intervention effects.

Finally, regarding the concern of a ceiling effect, in fact only 27.33% of participants chose the work option on all of 24 PEET trials. In comparison, for the WEPT task, the proportion at ceiling (8/8 trials) is 61.05%, significantly higher ($\chi^2_{(1)}$, $p<0.001$). Furthermore, unlike the WEPT, all our analyses on the PEET data compare between conditions (climate vs. food). This allows us to reveal intervention effects specific to climate motivation, relative to motivation to help the control cause.

We have included these points in an additional section on task validity in the Supplementary Information, expanding the section where we had already explored limitations of the WEPT that might explain the differences in our findings compared to those in Vlasceanu et al. (2024):

“In the model of choices that controlled for number of pages completed on the WEPT, we additionally examined the association between this control variable and willingness to choose effort in the PEET. We found that participants who completed more pages on the WEPT were also more willing to exert effort in the PEET (OR=1.73 [1.59, 1.88], $p<0.001$). This positive association between PEET and WEPT measures was unexpected given we found that several interventions increased willingness to work in the PEET but previous results reported that these interventions did not increase motivation to complete the WEPT, in a sample that included our participants¹. We therefore examined intervention effects on pages completed in the WEPT using cumulative link mixed models (CLMM; see Methods) in all participants in the ICPC sample from the six countries (of 63 in total) we collected data in. As previously reported¹, no interventions significantly increased pages completed in the WEPT and several had significant negative impacts: Psychological Distance, Negative Emotions, Work-Together Norm, and Letter to Future Generation (Table S5). Therefore, differences between the two studies in how interventions increased pro-environmental effort were not due to our sample being unrepresentative or underpowered. Finally, we probed whether intervention effects would be observed when only including participants who completed the WEPT and also met pre-registered exclusion criteria for analysis of the PEET. Here, five interventions significantly *increased* the number of pages completed in the WEPT: Pluralistic Ignorance, Scientific Consensus, Collective Action, Dynamic Social Norms, and Binding Moral Foundations (Table S5). This result highlights the importance of adequate checks for attention and choice options with participants in the PEET having to decide between working or resting. We also note that controlling for time spent in the study did not impact on intervention effects in the PEET

(Table S4), whereas controlling for the length of interventions also changed the effect of interventions on the WEPT results¹.

We additionally conducted several validity checks of the PEET to robustly establish the validity of our novel task. First, it could be argued that clicking boxes is not perceived as effortful or tiring but instead might be seen as a challenge or game. We therefore asked participants after the task to complete the NASA Task Load Index²² to test whether they perceived the highest effort level as significantly more effortful than the lowest and compared self-reports of tiredness after the PEET compared to before it (both ratings on 0-100 scales). Results showed that the highest effort level (mean rating=58.85) was perceived as significantly harder than the lowest effort level (mean=37.36, difference $d=0.81$ [0.77, 0.85], $p<0.001$). Similarly, participants were more tired at the end (mean rating=61.25) than before the start of the PEET (mean=49.67, difference $d=0.48$ [0.44, 0.52], $p<0.001$). Next, we considered the potential role of risk avoidance as an explanation for choosing rest over work. We compared the overall success rate after choosing to work in both conditions, as if success rates were low this would suggest risk was a factor driving participant choice. Instead, we observed high success rates in both conditions (climate mean=92.32%, food mean=91.54%) meaning when people did choose to work they were able to achieve the effort they chose. Finally, we examined the possibility of a ceiling effect in the PEET. This revealed 27.33% of participants chose the work option on all of 24 PEET trials. By comparison, for the WEPT task, the proportion at ceiling (8/8 pages completed) was significantly higher (61.05%, comparison $\chi^2_{(1)}$, $p<0.001$)."

We have also more explicitly compared the advantages of the PEET to limitations of existing measures in the revised manuscript:

Introduction

"More broadly, precisely controlled experimental designs are vital to understand willingness to engage in pro-environmental behaviors²⁶. Some existing measures of pro-environmental behaviour, such as the Work for Environmental Protection Task²⁷ (WEPT) and the Carbon Emission Task²⁸ (CET), have started to include aspects of experimental design including incentivising choices and using multiple trials. However, these tasks crucially lack a control condition. Previous work demonstrates the multidimensional nature of motivation – people can be apathetic to exert effort into some actions but not others. For example, they may be differentially willing to exert the same level of effort for the same amount of reward depending on who receives the reward^{15–17}. Effective paradigms therefore need a control cause, for example a non-environmental charity, to test whether people are broadly motivated or motivated only for specific recipients or causes. The need for a control condition is further emphasised when testing intervention effects. The fact that existing paradigms lack a control condition makes it impossible to test whether interventions increase motivation to help all causes or specifically the pro-environmental motivation they were designed to affect. This distinction is key for understanding the mechanisms of successful interventions and potential implications of applied use of interventions.

Extensive work on motivation in behavioural science and neuroscience highlights multiple additional important design aspects that tasks should incorporate^{9,10,14,22–24}. First, to understand the role of effort specifically, it is crucial to control for time. Decisions should be between options that take the same amount of time and differ only in the effort required, to ensure that choices are not made based on the temporal discounting of rewards²⁵. Second, the amount of effort required must be tailored to individuals' capacity for the effortful task, so differences between people are

driven by motivation, not skill. Third, the reward magnitude and the amount of effort required should vary independently to quantify the effects of each, as for some people incentives may matter more than effort or vice versa. Importantly, these strengths are not captured in existing measures such as the WEPT, which conflates cognitive and physical effort, time and reward. While the CET does vary pro-environmental benefits and financial cost to oneself independently, the effort costs of benefitting the climate are not manipulated. It is impossible to exclude the possibility that more pro-environmental behaviour is actually due to a lower subjective valuation of financial costs.”

Results

“Finally, we examined associations between pages completed in the WEPT and motivation to help each cause. WEPT performance was positively correlated with motivation for both the climate (negative correlation with K_s $r_{(3055)}=-0.16$ [-0.20, -0.13], $p<0.001$) and food ($r_{(3055)}=-0.18$ [-0.21, -0.14], $p<0.001$) charities, but importantly with no significant difference in the strength of these correlations ($Z=0.54$, $p=0.59$; Figure 5E). Together these results highlight that general motivation is an important factor in increasing willingness to exert effort for any cause, but the PEET captures the specificity of variability in motivation to help the climate driving individual differences in climate-relevant outcomes. This specificity was not true for the WEPT.”

Discussion

“Finally, establishing the validity of the PEET also highlights advantages relative to existing measures. While climate motivation measured by the PEET specifically linked to climate-relevant individual differences, the WEPT cannot show this specificity as it lacks a control condition. Performance on the WEPT was similarly associated with both climate and food motivation from the PEET, suggesting it may capture individuals’ general tendency to complete tasks rather than being climate specific. Our supplementary results also revealed that intervention effects on WEPT performance were not robust to controlling for time, whereas effects on PEET measures were, and that the WEPT showed a significantly higher ceiling effect (61% of participants) compared to the PEET (27% of participants). These findings emphasise the need for a new tightly controlled experimental measure of pro-environmental motivation.”

REVIEWER COMMENTS:

Reviewer #1 (Remarks to the Author):

I want to thank the authors for their detailed explanations and the thorough revision of their manuscript. While the contribution of the present work is now clearer to me, I continue to have some important concerns, in particular related to the results presentation and interpretation.

***Response:** Thank you for your time in providing further feedback on our manuscript. We are grateful for your positive comments on our thorough revisions, and your additional queries, which have helped further improve the manuscript.*

Major concerns

1) In my view, the manuscript would substantially benefit from a clearer choice between focusing on presenting the properties of the novel task and the analysis of its outcomes. The section “Environmental benefits are devalued when they require effort” is interesting for a reader that is interested in the validity of the design of the task, but does not provide any novel insights. I suggest removing this section from the main manuscript and more clearly focus on the analysis of the task outcomes. Related parts in the discussion, that seem obvious to a large audience of scholars, could then also be removed (e.g., “people were more willing to work for larger rewards, but less when the effort required was greater”).

***Response:** Thank you for your feedback. Our aims were indeed two-fold, to establish a novel task that overcomes some of the limitations of existing measures, and second, to test the effects of interventions on the new task. This two-fold aim is also directly in line with our pre-registered hypotheses. Indeed, the finding that environmental benefits are devalued when they require effort is novel. To our knowledge, this is the first time that a controlled experimental task has demonstrated the effect of increasing physical effort on devaluing pro-environmental rewards. While it might seem obvious based on previous work outside the environmental domain or from real-world observations of pro-environmental motivation, we think that demonstrating this in an experiment on pro-environmental effort for the first time is also important.*

However, we appreciate and agree that the balance of the manuscript needs to be clearer in terms of the two-aims and the conclusions that can be drawn from them, and we have adjusted the text in the introduction accordingly:

“First, we aimed to validate the PEET in a large international sample by testing whether motivation to help the environment varies with reward and effort, as has been shown in other domains. Establishing this for climate motivation in a controlled experimental task also presents an important step in the literature. Our second aim was to use this design to test the effect of 11 interventions on climate motivation.”

2) I have concerns with regard to the results for the choice outcomes presented in the section “Psychological interventions increase relative willingness to help the environment” for which

the hypothesized effects consistently fall right below the .05 alpha-level. In my view, this warrants extremely careful interpretation of these findings, and by no means claims about “robust evidence” (Figure 3).

Response: *We appreciate you raising these important points, which we have responded to one by one. We apologise for any confusion in our use of the term ‘robust’. We used this term in the manuscript to refer to interventions that have a significant effect on both choices and parameters capturing choice discounting from the computational model (Ks) across all control analyses. We have now changed the term to ‘consistent’ throughout the manuscript to avoid confusion.*

a. Do these p-values correspond to one-sided tests of the interaction coefficients?

Response: *These are two-sided tests. Although we hypothesized in our pre-registration that each intervention could increase relative climate motivation, we ensured our analysis could capture effects in either direction, making the test of each more stringent. We have clarified in the manuscript that the tests were two-sided:*

“We tested the impact of each intervention compared to the control group using a GLMM (two-sided tests”

b. Would corrections for multiple comparisons be warranted given the considerable amount of interventions tested?

Response: *Thank you for raising the important point about when it is appropriate to correct for multiple comparisons. Establishing when correcting for multiple comparisons is warranted, and when it is not, is key for balancing type I and type II errors. In this work, the nature of our hypotheses and interpretation means it is not appropriate to correct for multiple comparisons across the independent interventions.*

As outlined in the paper on this issue that Reviewer 2 mentioned (Rubin, 2021), there are three types of multiple comparison testing: “individual testing”, “conjunction testing”, and “disjunction testing”, and only disjunction testing requires correction. Applying the theoretical understanding and examples from this paper to our study gives the following application of these types:

Disjunction testing: a significant effect of one or more interventions is used to conclude that all interventions increase climate motivation.

Conjunction testing: the effect of all 11 interventions must be significant to conclude that interventions increase climate motivation

Individual testing: the effect of each intervention is considered an independent hypothesis (e.g. Psychological Distance intervention increases climate motivation) which is examined with exactly one test. Conclusions are limited to the effect of each intervention and not made across interventions.

All of our analysis, reporting of results, and summaries of results in the discussion reflect individual testing. For example, our mixed models test the interaction between the cause (food

/ climate) and each individual intervention (Tables S5, S8 & S9; tested with “summary(model)” in lme4) rather than the omnibus test (“Anova(model)”). Representative sections of the text in the results and discussion include [emphasis added here]:

“We hypothesized that at least some interventions would increase choices to exert effort and motivation to benefit the climate compared to the food charity, relative to the control group who read a brief narrative unrelated to climate change.”

“We tested the impact of each intervention compared to the control group using a GLMM”

“We next examined if any intervention increased relative motivation to help the environment. Four interventions significantly reduced the food over climate bias (Figure 4C)...”

“In summary, computational modelling revealed select interventions significantly changed discounting of reward by effort in pro-environmental choices”

“However, several interventions, particularly those that targeted psychological distance and system justification, significantly reduced this bias, meaning people were relatively more willing to exert effort into actions that benefit the climate.”

As stated by Rubin, such **individual testing does not require correction**:

“If a single test result is used to make a decision about a single null hypothesis [e.g. does Psychological Distance intervention have an effect, tested with the term in the model for the interaction between this intervention and cause], then that test result provides only one opportunity to make a Type I error about that null hypothesis. Consequently, the alpha level of the test (α Individual) does not need to be lowered. Importantly, the logic of individual testing applies even when multiple instances of individual testing take place side-by-side within the same study. If each decision to reject each individual null hypothesis depends on no more than one significance test, then none of the individual tests constitutes a “family” with respect to any single hypothesis. Consequently, it is not necessary to adjust alpha levels on the basis of any family-based error rate.”

This is also why the paper reporting the results of the ICPC study, which shows similar effect sizes to our work, does not provide a formal correction for multiple comparisons across the same 11 interventions and using a similar analysis approach (Vlasceanu et al. 2024, Science Advances).

However, importantly, we now recognise that in trying to summarise our findings concisely, in particular for the title of the manuscript, linguistically we had implied disjunction testing, which would have required correcting. This was not our intention, and we appreciate the opportunity to fix this oversight. We have made the following edits to the title of the manuscript, subheading within the results, and general text:

Psychological interventions that decrease psychological distance or challenge system justification increase motivation to exert effort to mitigate climate change

Three psychological interventions increase relative willingness to help the environment

We tested the effect of ~~these interventions~~ each intervention on two measures

We also note four aspects of the existing manuscript that support our results. First, the fact that the analysis of choices and Ks broadly agree, and we focus our conclusions only on where these results are consistent (“conjunction testing” across the two outcomes). Second, while the p values for the choices are closer to the threshold of 0.05 than the p values for Ks, the standardised effect sizes are actually larger for choices than Ks. The difference in p values is therefore due to the decreased standard error in Ks, likely due to the computational modelling accounting for differences in choice variability. This point highlights the fact that p values are determined by multiple factors, not simply the size of the effect. Third, we already included Bayesian tests, which do not rely on p values, to establish that following the interventions we conclude are successful, there is no difference in motivation between the climate and the food cause, whereas there is significant food bias in the control group. Finally, raw effect sizes are also interpretable independent of p values. A change of 3% following an intervention would have a huge impact if applied across everyday effortful pro-environmental behaviours and is similar to the effect sizes in previous published literature (e.g. Vlasceanu et al., 2024, Science Advances).

Based on the above, we can be confident that our analysis approach is indeed controlling the error rate at 0.05 for each hypothesis meaning we have a 5% chance of false positives, and the effect sizes for the interventions we report as significant are meaningful. We have now stated our significance criterion explicitly in the manuscript:

“We tested the impact of each intervention compared to the control group using a GLMM (two-sided tests, preregistered $\alpha=0.05$ applied to each individual intervention effect; see Methods).”

However, to further strengthen our results, we have now conducted new extensive simulation and bootstrapping analyses to further validate whether the effect sizes we see reflect real effects or are the result of error. In addition to supporting our significance testing, this allows us to build on the evidence from the effect size estimates, which are independent of p values, and demonstrate the precision of these.

*For this analysis, we simulated 1,000 datasets with the same structure and size as our real dataset. Without collecting more data, which would be impossible across multiple countries and authors, this is the best available way of establishing further substantive evidence for the strength of our results. We did this by sampling participant numbers with replacement the same number of times as the total number of participants (3,055), maintaining participants’ allocations to their intervention group. We did this for both choices and K parameters separately. On each simulated dataset, we then calculated the average difference between the control group and each intervention condition (raw % change for choices, raw difference in Ks) – corresponding to the averages plotted in Figure 3 and Figure 4C respectively. We also ran the relevant mixed model on the simulated dataset and extracted the standardised effect size (odds ratio) for the cause * intervention interactions. These correspond to the model terms, with their associated p values, that we use for our conclusions. For each intervention and each outcome measure (choices and Ks), this generates 1000 samples with two effect size metrics: one raw and one from the model. These are shown for System Justification on choices as an example below.*

Our simulated datasets allowed us to establish two aspects of robustness. First, it can **support the conclusions of our significance testing**. If significant results were due to noise in our sample, these simulations would show distributions of effects which include the null hypothesis of no effect. In contrast, if the 95th percentile of the distributions do not include the null, this is evidence for a genuine effect. Second, we can establish the **precision of our effect size estimates**, which are **independent of p values**. If our reported effect size estimates were spuriously large, leading to false positives, these would fall in the upper tail of the simulated distribution. If our reported effect sizes are close to the median of the distribution this provides reassurance that these effects are less prone to change. The ranges of these distributions are also informative for establishing the likely effect sizes of all interventions for future work.

In the example above, on the left histogram, the 95th percentile is 0.56%, which is above zero, meaning the null hypothesis of no difference is not included. For the effect size from the model (right histogram), the value for the real sample was 0.86, and as these are odds ratios, a value of 1 represents no effect. Here the 95th percentile is 0.97, so again, not including the odds ratio that would indicate no effect. These findings support the robustness of our inference using p values. For the raw % change in choices to help (left histogram) the average effect in our real sample was approximately 3% and this is well replicated by the median of the sampled distribution, demonstrating our effect size estimate is precise.

For all of the significant effects in the paper, as determined by the reported p value, across choices and Ks, it was the case that the distributions (95th percentile) of our 1,000 simulated datasets did not include the value corresponding to no effect. This was true for both the analysis of raw differences (null = 0, alternative hypothesis >0) and model odds ratios (null = 1, alternative hypothesis for choices <1, Supplementary Table 7; for Ks >1, Supplementary Table 11). In contrast, the interventions that did not show a significant effect did include the null value in their distributions (see tables below). In all cases, the medians of the simulated distributions closely reflected the effect sizes reported in the main manuscript (Figure S3 & Figure S4). Taken together, these results show the sensitivity, specificity, and precision of our

existing analysis and allow us to have greater confidence that the significant effects we observe are genuine effects. We have added the results of this new simulation analysis to the to the supplementary materials and signposted to them in the main manuscript:

Results

“Effect sizes for the interventions with significant effects showed approximately 3% increases in choices for climate relative to food (PD=3.55%, SJ=2.79%, NE=3.08%; Figure 3). To further strengthen the significance testing results and capture the precision of these effect size estimates we additionally conducted bootstrapping analysis with 1,000 simulations of the dataset (see Supplementary Methods). Results showed that the medians of the simulation distributions closely captured our reported effects, providing confidence in their precision. Supporting the results of the interventions with significant effects in our main GLMM, the 95th percentiles of the simulated distributions did not include the value corresponding to no effect. In contrast, distributions (95th percentile) for the non-significant interventions included the null value (see Supplementary Results, Figure S3, and Table S7).”

“Finally, as with the choices, bootstrapping analysis provided further evidence for these results and the precision of the effect sizes (see Supplementary Results, Figure S4, and Table S11).”

Supplementary Methods

Bootstrapping analysis

We simulated 1,000 datasets with the same structure and size as our real dataset. Without collecting more data, which would be impossible across multiple countries and authors, this is the best available way of establishing further substantive evidence for the strength of our results. We did this by sampling participant numbers with replacement the same number of times as the total number of participants (3,055), maintaining participants' allocations to their intervention group. We did this for both choices and *K* parameters separately. On each simulated dataset, we then calculated the average difference between the control group and each intervention condition (raw % change for choices, raw difference in *Ks*) – corresponding to the averages plotted in Figure 3 and Figure 4C respectively. We also ran the relevant mixed model on the simulated dataset and extracted the standardised effect size (odds ratio) for the cause * intervention interactions. These correspond to the model terms, with their associated p values, that we use for our conclusions. For each intervention and each outcome measure (choices and *Ks*), this generates 1000 samples with two effect size metrics: one raw and one from the model.

Supplementary Results

Bootstrapping analysis

For all of the significant effects in the paper, as determined by the reported p value, across choices and K_s , it was the case that the distributions (95th percentile) of our 1,000 simulated datasets did not include the value corresponding to no effect. This was true for both the analysis of raw differences (null = 0, alternative hypothesis >0) and model odds ratios (null = 1, alternative hypothesis for choices <1 , Supplementary Table 7; for $K_s >1$, Supplementary Table 11). In contrast, the interventions that did not show a significant effect did include the null value in their distributions (see tables below). In all cases, the medians of the simulated distributions closely reflected the effect sizes reported in the main manuscript (Figure S3 & Figure S4). Taken together, these results show the sensitivity, specificity, and precision of our existing analysis and allow us to have greater confidence that the significant effects we observe are genuine effects.

Figure S3. Simulation analysis of choices supporting effect size of successful interventions. Boxplot shows the median, lower and upper quartiles, and 5% and 95% percentiles of the simulated distributions. The X shows the effect size from the main dataset (as in Figure 3). The medians of the simulations closely capture the effect sizes in all cases. The 95% percentile does not cross 0 for the interventions that have a significant effect on choices. The upper and lower quartiles for all interventions, both significant and non-significant, could be useful for future research to provide the confidence around an expected effect size.

Table S7: Results of simulation analysis for choices

Parameter	OR	SE	CI low	CI high	z	p	Raw 95%	Model 95%
Cause (Food > Climate) * Work Together Norm	0.92	0.07	0.79	1.07	-1.10	0.27	-0.88	1.05
Cause (Food > Climate) * Negative Emotions	0.85	0.06	0.74	0.99	-2.10	0.036	0.67	0.97
Cause (Food > Climate) * Scientific Consensus	0.92	0.07	0.79	1.06	-1.17	0.24	-0.19	1.02
Cause (Food > Climate) * Effective Collective Action	0.93	0.07	0.80	1.07	-1.02	0.31	-1.01	1.05
Cause (Food > Climate) * System Justification	0.86	0.06	0.74	1.00	-2.01	0.044	0.56	0.97
Cause (Food > Climate) * Psychological Distance	0.82	0.06	0.71	0.96	-2.51	0.012	1.38	0.93
Cause (Food > Climate) * Pluralistic Ignorance	0.86	0.07	0.74	1.00	-1.96	0.0498	0.64	0.97
Cause (Food > Climate) * Letter to Future Generation	1.02	0.08	0.86	1.19	0.18	0.86	-2	1.16
Cause (Food > Climate) * Dynamic Social Norms	0.93	0.07	0.80	1.08	-0.95	0.34	-0.89	1.07
Cause (Food > Climate) * Future-Self Continuity	0.85	0.07	0.72	1.00	-1.92	0.055	0.65	0.98
Cause (Food > Climate) * Binding Moral Foundations	0.92	0.07	0.79	1.07	-1.02	0.31	-0.68	1.05

Note. OR: odds ratio, SE: standard error of the mean, CI: confidence interval. Columns up to *p* duplicate those from Table S5 with rows that do not correspond to interaction effects not shown for brevity. The final two columns give the results of the simulation analyses. Effect 95% corresponds to the 95% percentile of the distribution of raw differences in choice % for climate vs. food in 1,000 simulated datasets. The null hypothesis here is 0 and the effect in the real data is positive. Interventions with a significant effect are shown in bold italic and none of the 95th percentile values cross 0. Model 95% provides the 95% percentile of the distribution in odds ratios for the mixed model of choices in 1,000 simulated datasets. The null hypothesis here is 1 and the effect in the real data is below 1 (see first OR column). None of the interventions with a significant effect cross 1.

Figure S4. Simulation analysis of choices supporting effect size of successful interventions. Boxplot shows the median, lower and upper quartiles, and 5% and 95% percentiles of the simulated distributions. The X shows the effect size from the main dataset (as in Figure 4c). The medians of the simulations closely capture the effect sizes in all cases. The 95% percentile does not cross 0 for the interventions that have a significant effect on choices. The upper and lower quartiles for all interventions, both significant and non-significant, could be useful for future research to provide the confidence around an expected effect size.

Table S11: Results of simulation analysis for K parameters

Parameter	OR	SE	CI low	CI high	$t_{(6082)}$	p	Raw 95%	Model 95%
Cause (Food > Climate) * Work Together Norm	1.02	0.02	0.99	1.06	1.27	0.2	-0.004	0.968
Cause (Food > Climate) * Negative Emotions	1.04	0.02	1	1.07	1.95	0.051	0.004	0.997
Cause (Food > Climate) * Scientific Consensus	1.01	0.02	0.98	1.05	0.69	0.49	-0.001	0.954
Cause (Food > Climate) * Effective Collective Action	1.02	0.02	0.99	1.06	1.12	0.26	-0.012	0.964
Cause (Food > Climate) * System Justification	1.05	0.02	1.01	1.08	2.64	0.008	0.009	1.019
Cause (Food > Climate) * Psychological Distance	1.06	0.02	1.03	1.1	3.42	<0.001	0.015	1.048
Cause (Food > Climate) * Pluralistic Ignorance	1.04	0.02	1.01	1.08	2.3	0.022	0.005	1.004
Cause (Food > Climate) * Letter to Future Generation	1.01	0.02	0.97	1.05	0.5	0.61	-0.004	0.946
Cause (Food > Climate) * Dynamic Social Norms	1.02	0.02	0.99	1.06	1.38	0.17	-0.009	0.977

Cause (Food > Climate) *								
Future-Self Continuity	1.04	0.02	1	1.08	2.22	0.027	0.003	1.005
Cause (Food > Climate) *								
Binding Moral Foundations	1.02	0.02	0.98	1.05	0.92	0.36	-0.009	0.959

Note. OR: odds ratio, SE: standard error of the mean, CI: confidence interval. Columns up to *p* duplicate those from Table S10 with rows that do not correspond to interaction effects not shown for brevity. The final two columns give the results of the simulation analyses. Effect 95% corresponds to the 95th percentile of the distribution of raw differences in *Ks* for climate vs. food in 1,000 simulated datasets. The null hypothesis here is 0 and the effect in the real data is positive. Interventions with a significant effect are shown in bold italic and none of the 95th percentile values cross 0. Model 95% provides the 95% percentile of the distribution in odds ratios for the mixed model of choices in 1,000 simulated datasets. The null hypothesis here is 1 and the effect in the real data is above 1 (see first OR column). None of the interventions with a significant effect cross 1.

c. How does the selected random structure influence the results?

Response: Thank you for your query. We took a principled approach to selecting our random structure, following our preregistered analysis plan and the guidance of Matushchek et al. (2017). We fit maximal models to the data, then reduced the maximal model to achieve convergence and maximise power, first fitting a zero-correlation model with all random effects and interactions but without correlations, then removing interactions in the random effects, starting with those that explained the lowest amounts of variance. We did this first in the simpler case of the main effects of effort, reward, and climate vs. food on choices, and then used the reduced model identified here as a starting point to add the intervention factor (12 levels) to. We did then try adding random effects of intervention, but this model could not fit. **The model with the largest random effects structure for the effect of the interventions on choices that did finish fitting, but with singular fit non-convergence issues, showed significant intervention effects of the same four interventions identified in the main analysis (Negative Emotions, System Justification, Psychological Distance and Pluralistic Ignorance).** While this model could not be used for conclusions alone due to the convergence issues, it shows that the results are the same with different random effects structures. Fortunately, the nature of the *K* data means the space of possible random structures is significantly reduced and the maximum possible model fit without any issues. Therefore, as also highlighted above, the correspondence in significant intervention effects between these two measures supports the results of the choice analysis on this issue of random effects, too. Finally, we are transparent in our selection approach, keeping our commented steps in the shared code and also sharing the output files of all fitted models on OSF (<https://osf.io/zv2tu/files/01a86854-79fe-4d7e-aed0-7286123d28e3#> in Github / PM_R_code / output). We have also added equations for the models, including the random effects, to the manuscript:

“Binomial GLMMs predicting people’s decision to accept the high-effort high-reward work offer included within-subject fixed effects of reward available (level 2-4: 4, 12, 20 credits), effort required (level 2-5: 50, 65, 80, 95% maximum), and cause (climate vs. food). Random effects were grouped by participant nested in country and removed when necessary to obtain a converging model that maximizes power while minimizing Type I errors⁶³. A fixed between-subject effect of intervention group and interaction between intervention and cause (climate / food) was then added to this model, making the final model:

choice ~ effort + reward + cause*intervention + (0 + effort + reward | country/participant) + (1 + agent | participant:country)

In my view, it would be appropriate to tone down any claims about the effects of the interventions on the choice outcome, and more strongly focus on the modeling results which appear more robust. The choice outcome results could for example be moved behind the modeling results. Unfortunately, I was not able to run the .Rproject shared through osf/git to verify some of the questions raised above.

***Response:** We hope our responses to your previous queries above have outlined more clearly the strength of our results. We appreciate the suggestion to reorder the results. The reason we ordered the results to start with the choices and then with the modelling is to make sure the results are accessible, as not all readers will have experience with computational modelling. We believe that this is a sensible approach and that prioritising the ordering based on p values, could be problematic as these are not necessarily interpretable on their own for the reasons outlined above. We have however expanded the existing point in the discussion regarding the effect size of interventions to include these for both choices and Ks:*

“The interventions effects were also relatively small. Considering the raw difference in the percentage of times participants chose to work showed a 3% relative increase for the climate. Odds ratios from the models showed the effect of interventions on the odds of choices is 16-22% greater in the climate condition compared to the food condition for the successful interventions. For the odds of Ks this ranged from 4-6%. However, small changes to the everyday behaviours of millions of people make a sizable difference”

Regarding the running of our code, we appreciate you taking the time to try and access our resources and apologies for the inconvenience. We have verified the code through two colleagues who are not associated with the project downloading and testing it. On some computers, there was an issue with printing the figures to the html output when knitting that caused an error. This has now been fixed, although some users may still need to install or update packages to run the code, all of these are included within the script. From what our colleagues reported, it was always possible to access the code itself to see that the tests were two-sided and where we record our model selection approach, as well as load in the data and saved model output. If there are still issues for you, please let us know further details so we can look into this.

3) The integration of the present work into the bigger survey on climate outcomes and including the WEPT is potentially problematic. It's a very particular setup to test a novel task after participants have spent about 20 minutes completing loads of other measures related to climate change and the environment. In particular, I wonder whether this might have emphasized consistent (pro-environmental) responding in the intervention groups and demand effects increasing the gap between relative preference for the climate over the food charity. This very particular circumstance should be more transparently discussed and its implications on the theoretical and practical implications that can be drawn from the collected data.

Response: Thank you for this consideration about the wider context of our study. We agree that further elaboration here is important. We appreciate that participants completed several measures related to climate change before our task. However, this is not unusual scientifically and is the case in many research projects, particularly those wanting to maximise the impact of recruiting and testing such large international samples. Fortunately, our pattern of results is highly unlikely to be explained by this feature of the study, as the control group also completed all the same measures related to climate change before completing the PEET. These participants showed a significant bias towards helping the food charity over the climate cause. Participants in other intervention conditions that we do not report as successful similarly completed all the same measures. Therefore, participants who experienced the successful interventions, particularly Psychological Distance and System Justification that had consistent significant effects across analyses, were not subject to any different demand effects than the other groups but were equally willing to help the climate as the food cause. We have now more transparently highlighted this aspect of our study in the discussion and explained for the above reasons why the implications on our conclusions are minimal:

“Future research should also test participants on the PEET without this following other measures of pro-environmental attitudes and intentions. However, any demand characteristics that could be driven by the context of this study cannot explain results as the control group were matched to the intervention groups on completing all of these measures.”

Minor concerns

4) Given that other parts might move to the supplementary material, I still think that more information on the content of the interventions is needed in the main manuscript.

Response: Thank you for this suggestion, we have added the following information to the main manuscript:

Introduction:

“Participants were randomly assigned to one of 12 groups, either a control group or one of the 11 pro-environmental psychological interventions developed by experts from the International Climate Psychology Collaboration (ICPC)^{3,4} based on empirical and theoretical work. Each intervention used some or all of images, text and asking participants to enter text (see Methods), based on the theme and previous research supporting it. We tested the effect [...]

“However, two interventions consistently increased relative motivation to help the environment. The first, Psychological Distance, highlighted climate change as an immediate, local threat using two examples of recent natural disasters in the participants country and participants reflected on how it affects them personally through selecting ways from a list and a free text response. The second, System Justification, used text and images to emphasize the importance of the environment to the participants’ life, current consequences of climate change such as weather events that affect them and other people in their country, and an appeal to patriotic values as reasons to preserve the environment. Overall,”

Methods:

Interventions

Working-together norms. Participants read a flier promoting climate action as a collective effort, reinforcing the idea of working together with others to reduce carbon emissions.

System justification. Text and images framed climate change as a threat to participants' way of life and encouraged pro-environmental behaviour as patriotic.

Binding moral foundations. Participants read a message invoking national pride, loyalty, and authority to support clean energy and climate action.

Exposure to effective collective action. Participants were shown examples of successful climate-related movements to inspire hope and belief in the power of collective action.

Future self-continuity. Participants imagined a future version of themselves and wrote a letter to their present self about the importance of taking climate action now.

Scientific consensus. Participants saw a message and graphic emphasizing that 99% of climate scientists agree climate change is real and caused by humans.

Decreasing psychological distance. Climate change was presented as an immediate, local threat, and participants reflected on how it affects them personally.

Dynamic social norms. Participants read that more people are taking climate action globally over time, supported by examples and data showing behavioural trends.

Correcting pluralistic ignorance. Participants were shown how concern about climate change is much more widespread than people typically believe.

Letter to future generations. Participants wrote a letter to a future child or other family member, describing their efforts to protect the planet and how they wish to be remembered.

Negative emotion. Participants were exposed to emotionally intense, alarming climate information designed to induce negative emotions.

Control group. Participants read a neutral passage of text not related to climate change from *Great Expectations*.

5) The internal consistency within the used sample (and not the many labs) should be presented for the climate beliefs and climate policy support measures.

Response: Thank you for this suggestion, we have added the internal consistency for the current sample and note it is almost identical to the large ICPC sample:

Climate beliefs³. [...] The measure had high internal consistency in the large ICPC sample^{3,4} (Cronbach's alpha=0.93, n=59,440) and in the participants included in our analysis (Cronbach's alpha=0.94, n=3,055).

Climate policy support³. [...] The internal consistency in the large ICPC sample^{3,4} and the sample presented here was high (Cronbach's alpha=0.88, n=59,440; Cronbach's alpha=0.89, n=3,055).

6) Use consistent terms for climate and food charity (not "control condition") from the start of the paper.

Response: We have edited this throughout the manuscript.

7) The terms "precisely" and "specifically" are used way too often throughout the manuscript and obstruct fluent reading without adding information or increasing the convincingness of the presented argumentation or evidence. I suggest to remove at least half of them.

Response: We have implemented this suggestion.

8) Provide exact p value instead of $p < .05$ in footnote of Figure 2.

Response: This section of the legend now reads "The asterisk between food and climate represents a significant effect ($p=0.017$) of cause in the GLMM of choices.

9) "Strikingly, control participants showed a significant bias towards helping the food charity, over the climate charity (OR=1.13 [1.02, 1.25], $p=0.017$;" I don't think that "striking" is the right word for this exploratory finding which just falls below the magic .05 alpha-level.

Response: We have removed the word 'strikingly'.

10) For all figures, display error bars for 95% confidence intervals instead of standard errors.

Response: Thank you for this suggestion. We already plot 95% CIs for continuous effects on line graphs with shaded error areas. Our error bars on bar graphs comply with the Nature Portfolio Editorial Policy that states that standard deviation, standard error or confidence intervals can be used as long as this is clear in the figure legend.

11) The authors should acknowledge that there are also high impact climate actions that do not require more effort.

Response: Thank you. We have edited the discussion as follows. We also note that throughout we state pro-environmental actions are "often" rather than "always" more effortful.

In addition to the strengths of the work, it is important to recognize limitations. We used a tightly controlled experimental task to precisely isolate and quantify the role of motivation to exert effort, but this may limit the external validity of the findings. Combining complementary approaches using experiments with a variety of real-world pro-environmental behaviours will be vital to fully understand how people choose to help mitigate climate change. This should include both actions that require more effort and high impact climate actions that do not require more effort than the alternative.

Reviewer #2 (Remarks to the Author):

I thank the authors for their comprehensive responses to my queries about the added value, which I think are now adequately addressed. I still have key questions about the statistical analyses.

Response: *Thank you for highlighting our previous responses were comprehensive and for your additional feedback. Our new revisions have helped us further strengthen the manuscript and we are grateful for your time.*

First, the authors did not correct for multiple testing. Correcting for multiple testing is appropriate if one (or more) significant effects out of the set is sufficient to reject the overall null hypothesis (e.g., Rubin, 2021). I think this applies here since the authors are rejecting their overall null hypothesis (i.e., interventions do not affect the relative effort exerted on the PEET) if just one of the interventions had a significant effect. In other words, the authors had 11 ‘opportunities’ to reject the null hypothesis (see Rubin, 2021, p. 10978). Importantly, if the alpha level is adjusted to account for multiple testing, this renders all main findings in this paper non-significant ($0.05 / 11 =$ a significance threshold of 0.005). I am personally of the opinion that correcting for multiple testing would be appropriate here, also given the fact that the significant results reported by the authors float around 0.05, which makes me question their robustness. I would like to see a reflection from the authors on this.

Response: *Thank you for raising the important point about whether it is appropriate to correct for multiple comparisons, it has highlighted an oversight in the wording of our title which we are grateful for the opportunity to fix. Throughout the paper, our hypotheses and interpretation reflect “individual testing” in Rubin’s terminology. We had 11 independent hypotheses, one for each intervention that it would increase climate motivation. We therefore have exactly one test of each hypothesis and we interpret the findings at the individual intervention level throughout the manuscript. For example, our mixed models test the interaction between the cause (food / climate) and each individual intervention (Tables S5, S8 & S9; tested with “summary(model)” in lme4) rather than the omnibus test (“Anova(model)”). Representative sections of the text in the results and discussion include [emphasis added here]:*

“We hypothesized that at least some interventions would increase choices to exert effort and motivation to benefit the climate compared to the food charity, relative to the control group who read a brief narrative unrelated to climate change.”

“We next examined if any intervention increased relative motivation to help the environment. Four interventions significantly reduced the food over climate bias (Figure 4C)...”

“In summary, computational modelling revealed select interventions significantly changed discounting of reward by effort in pro-environmental choices”

“However, several interventions, particularly those that targeted psychological distance and system justification, significantly reduced this bias, meaning people were relatively more willing to exert effort into actions that benefit the climate.”

*As stated by Rubin, such **individual testing does not require correction**:*

“If a single test result is used to make a decision about a single null hypothesis [e.g. does Psychological Distance intervention have an effect, tested with the term in the model for the

interaction between this intervention and cause], then that test result provides only one opportunity to make a Type I error about that null hypothesis. Consequently, the alpha level of the test (α individual) does not need to be lowered. Importantly, the logic of individual testing applies even when multiple instances of individual testing take place side-by-side within the same study. If each decision to reject each individual null hypothesis depends on no more than one significance test, then none of the individual tests constitutes a “family” with respect to any single hypothesis. Consequently, it is not necessary to adjust alpha levels on the basis of any family-based error rate.”

This is also why the paper reporting the results of the ICPC study, which shows similar effect sizes to our work, does not correct for multiple comparisons across the same 11 interventions and similar analysis approach (Vlasceanu et al. 2024, Science Advances).

However, importantly, we now recognise that in trying to summarise our findings concisely, in particular for the title of the manuscript, linguistically we have implied disjunction testing which would have required correcting. This was not our intention, and we appreciate the opportunity to fix this oversight. We have made the following edits to the title of the manuscript, subheading within the results, and general text:

Psychological interventions that decrease psychological distance or challenge system justification increase motivation to exert effort to mitigate climate change

Three psychological interventions increase relative willingness to help the environment

We tested the effect of ~~these interventions~~ each intervention on two measures

We also note four aspects of the existing manuscript that support our results. First, the fact that the analysis of choices and Ks broadly agree, and we focus our conclusions only on where these results are consistent (“conjunction testing” across the two outcomes). Second, while the p values for the choices are closer to the threshold of 0.05 than the p values for Ks, the standardised effect sizes are actually larger for choices than Ks. The difference in p values is therefore due to the decreased standard error in Ks, likely due to the computational modelling accounting for differences in choice variability. This point highlights the fact that p values are determined by multiple factors, not simply the size of the effect. Third, we already included Bayesian tests, which do not rely on p values, to establish that following the interventions we conclude are successful, there is no difference in motivation between the climate and the food cause, whereas there is significant food bias in the control group. Finally, raw effect sizes are also interpretable independent of p values. A change of 3% following an intervention would have a huge impact if applied across everyday effortful pro-environmental behaviours and is similar to the effect sizes in previous published literature (e.g. Vlasceanu et al., 2024, Science Advances).

Based on the above, we can be confident that our analysis approach is indeed controlling the error rate at 0.05 for each hypothesis meaning we have a 5% chance of false positives, and the effect sizes for the interventions we report as significant are meaningful. We have now stated our significance criterion explicitly in the manuscript:

“We tested the impact of each intervention compared to the control group using a GLMM (two-sided tests, preregistered alpha=0.05 applied to each individual intervention effect; see Methods).”

However, to further strengthen our results, we have now conducted new extensive simulation and bootstrapping analyses to further validate whether the effect sizes we see reflect real effects or are the result of error. In addition to supporting our significance testing, this allows us to build on the evidence from the effect size estimates, which are independent of p values, and demonstrate the precision of these.

*For this analysis, we simulated 1,000 datasets with the same structure and size as our real dataset. Without collecting more data, which would be impossible across multiple countries and authors, this is the best available way of establishing further substantive evidence for the strength of our results. We did this by sampling participant numbers with replacement the same number of times as the total number of participants (3,055), maintaining participants' allocations to their intervention group. We did this for both choices and K parameters separately. On each simulated dataset, we then calculated the average difference between the control group and each intervention condition (raw % change for choices, raw difference in Ks) – corresponding to the averages plotted in Figure 3 and Figure 4C respectively. We also ran the relevant mixed model on the simulated dataset and extracted the standardised effect size (odds ratio) for the cause * intervention interactions. These correspond to the model terms, with their associated p values, that we use for our conclusions. For each intervention and each outcome measure (choices and Ks), this generates 1000 samples with two effect size metrics: one raw and one from the model. These are shown for System Justification on choices as an example below.*

*Our simulated datasets allowed us to establish two aspects of robustness. First, it can **support the conclusions of our significance testing**. If significant results were due to noise in our sample, these simulations would show distributions of effects which include the null hypothesis of no effect. In contrast, if the 95th percentile of the distributions do not include the null, this is*

evidence for a genuine effect. Second, we can establish the **precision of our effect size estimates**, which are **independent of p values**. If our reported effect size estimates were spuriously large, leading to false positives, these would fall in the upper tail of the simulated distribution. If our reported effect sizes are close to the median of the distribution this provides reassurance that these effects are less prone to change. The ranges of these distributions are also informative for establishing the likely effect sizes of all interventions for future work.

In the example above, on the left histogram, the 95th percentile is 0.56%, which, crucially, is above zero, meaning the null hypothesis of no difference is not included. For the effect size from the model (right histogram), the value for the real sample was 0.86, and as these are odds ratios, a value of 1 represents no effect. Here the 95th percentile is 0.97, so again, not including the odds ratio that would indicate no effect. These findings support the robustness of our inference using p values. For the raw % change in choices to help (left histogram) the average effect in our real sample was approximately 3% and this is well replicated by the median of the sampled distribution, demonstrating our effect size estimate is precise.

For all of the significant effects in the paper, as determined by the reported p value, across choices and Ks, it was the case that the distributions (95th percentile) of our 1,000 simulated datasets did not include the value corresponding to no effect. This was true for both the analysis of raw differences (null = 0, alternative hypothesis >0) and model odds ratios (null = 1, alternative hypothesis for choices <1, Supplementary Table 7; for Ks >1, Supplementary Table 11). In contrast, the interventions that did not show a significant effect did include the null value in their distributions (see tables below). In all cases, the medians of the simulated distributions closely reflected the effect sizes reported in the main manuscript (Figure S3 & Figure S4). Taken together, these results show the sensitivity, specificity, and precision of our existing analysis and allow us to have greater confidence that the significant effects we observe are genuine effects. We have added the results of this new simulation analysis to the supplementary materials and signposted to them in the main manuscript:

Results

“Effect sizes for the interventions with significant effects showed approximately 3% increases in choices for climate relative to food (PD=3.55%, SJ=2.79%, NE=3.08%; Figure 3). To further strengthen the significance testing results and capture the precision of these effect size estimates we additionally conducted bootstrapping analysis with 1,000 simulations of the dataset (see Supplementary Methods). Results showed that the medians of the simulation distributions closely captured our reported effects, providing confidence in their precision. Supporting the results of the interventions with significant effects in our main GLMM, the 95th percentiles of the simulated distributions did not include the value corresponding to no effect. In contrast, distributions (95th percentile) for the non-significant interventions included the null value (see Supplementary Results, Figure S3, and Table S7).”

“Finally, as with the choices, bootstrapping analysis provided further evidence for these results and the precision of the effect sizes (see Supplementary Results, Figure S4, and Table S11).”

Supplementary Methods

Bootstrapping analysis

We simulated 1,000 datasets with the same structure and size as our real dataset. Without collecting more data, which would be impossible across multiple countries and authors, this is the best available way of establishing further substantive evidence for the strength of our results. We did this by sampling participant numbers with replacement the same number of times as the total number of participants (3,055), maintaining participants' allocations to their intervention group. We did this for both choices and K parameters separately. On each simulated dataset, we then calculated the average difference between the control group and each intervention condition (raw % change for choices, raw difference in K s) – corresponding to the averages plotted in Figure 3 and Figure 4C respectively. We also ran the relevant mixed model on the simulated dataset and extracted the standardised effect size (odds ratio) for the cause * intervention interactions. These correspond to the model terms, with their associated p values, that we use for our conclusions. For each intervention and each outcome measure (choices and K s), this generates 1000 samples with two effect size metrics: one raw and one from the model.

Supplementary Results

Bootstrapping analysis

For all of the significant effects in the paper, as determined by the reported p value, across choices and K s, it was the case that the distributions (95th percentile) of our 1,000 simulated datasets did not include the value corresponding to no effect. This was true for both the analysis of raw differences (null = 0, alternative hypothesis >0) and model odds ratios (null = 1, alternative hypothesis for choices <1 , Supplementary Table 7; for K s >1 , Supplementary Table 11). In contrast, the interventions that did not show a significant effect did include the null value in their distributions (see tables below). In all cases, the medians of the simulated distributions closely reflected the effect sizes reported in the main manuscript (Figure S3 & Figure S4). Taken together, these results show the sensitivity, specificity, and precision of our existing analysis and allow us to have greater confidence that the significant effects we observe are genuine effects.

Figure S3. Simulation analysis of choices supporting effect size of successful interventions. Boxplot shows the median, lower and upper quartiles, and 5% and 95% percentiles of the simulated distributions. The X shows the effect size from the main dataset (as in Figure 3). The medians of the simulations closely capture the effect sizes in all cases. The 95% percentile does not cross 0 for the interventions that have a significant effect on choices. The upper and lower quartiles for all interventions, both significant and non-significant, could be useful for future research to provide the confidence around an expected effect size.

Table S7: Results of simulation analysis for choices

Parameter	OR	SE	CI low	CI high	z	p	Raw 95%	Model 95%
Cause (Food > Climate) * Work Together Norm	0.92	0.07	0.79	1.07	-1.10	0.27	-0.88	1.05
Cause (Food > Climate) * Negative Emotions	0.85	0.06	0.74	0.99	-2.10	0.036	0.67	0.97
Cause (Food > Climate) * Scientific Consensus	0.92	0.07	0.79	1.06	-1.17	0.24	-0.19	1.02
Cause (Food > Climate) * Effective Collective Action	0.93	0.07	0.80	1.07	-1.02	0.31	-1.01	1.05
Cause (Food > Climate) * System Justification	0.86	0.06	0.74	1.00	-2.01	0.044	0.56	0.97
Cause (Food > Climate) * Psychological Distance	0.82	0.06	0.71	0.96	-2.51	0.012	1.38	0.93
Cause (Food > Climate) * Pluralistic Ignorance	0.86	0.07	0.74	1.00	-1.96	0.0498	0.64	0.97
Cause (Food > Climate) * Letter to Future Generation	1.02	0.08	0.86	1.19	0.18	0.86	-2	1.16
Cause (Food > Climate) * Dynamic Social Norms	0.93	0.07	0.80	1.08	-0.95	0.34	-0.89	1.07

Cause (Food > Climate) * Future-Self Continuity	0.85	0.07	0.72	1.00	-1.92	0.055	0.65	0.98
Cause (Food > Climate) * Binding Moral Foundations	0.92	0.07	0.79	1.07	-1.02	0.31	-0.68	1.05

Note. OR: odds ratio, SE: standard error of the mean, CI: confidence interval. Columns up to p duplicate those from Table S5 with rows that do not correspond to interaction effects not shown for brevity. The final two columns give the results of the simulation analyses. Effect 95% corresponds to the 95% percentile of the distribution of raw differences in choice % for climate vs. food in 1,000 simulated datasets. The null hypothesis here is 0 and the effect in the real data is positive. Interventions with a significant effect are shown in bold italic and none of the 95th percentile values cross 0. Model 95% provides the 95% percentile of the distribution in odds ratios for the mixed model of choices in 1,000 simulated datasets. The null hypothesis here is 1 and the effect in the real data is below 1 (see first OR column). None of the interventions with a significant effect cross 1.

Figure S4. Simulation analysis of choices supporting effect size of successful interventions. Boxplot shows the median, lower and upper quartiles, and 5% and 95% percentiles of the simulated distributions. The X shows the effect size from the main dataset (as in Figure 4c). The medians of the simulations closely capture the effect sizes in all cases. The 95% percentile does not cross 0 for the interventions that have a significant effect on choices. The upper and lower quartiles for all interventions, both significant and non-significant, could be useful for future research to provide the confidence around an expected effect size.

Table S11: Results of simulation analysis for K parameters

Parameter	OR	SE	CI low	CI high	$t_{(6082)}$	p	Raw 95%	Model 95%
Cause (Food > Climate) * Work Together Norm	1.02	0.02	0.99	1.06	1.27	0.2	-0.004	0.968
Cause (Food > Climate) * Negative Emotions	1.04	0.02	1	1.07	1.95	0.051	0.004	0.997

Cause (Food > Climate) *									
Scientific Consensus	1.01	0.02	0.98	1.05	0.69	0.49	-0.001	0.954	
Cause (Food > Climate) *									
Effective Collective Action	1.02	0.02	0.99	1.06	1.12	0.26	-0.012	0.964	
Cause (Food > Climate) *									
System Justification	1.05	0.02	1.01	1.08	2.64	0.008	0.009	1.019	
Cause (Food > Climate) *									
Psychological Distance	1.06	0.02	1.03	1.1	3.42	<0.001	0.015	1.048	
Cause (Food > Climate) *									
Pluralistic Ignorance	1.04	0.02	1.01	1.08	2.3	0.022	0.005	1.004	
Cause (Food > Climate) * Letter									
to Future Generation	1.01	0.02	0.97	1.05	0.5	0.61	-0.004	0.946	
Cause (Food > Climate) *									
Dynamic Social Norms	1.02	0.02	0.99	1.06	1.38	0.17	-0.009	0.977	
Cause (Food > Climate) *									
Future-Self Continuity	1.04	0.02	1	1.08	2.22	0.027	0.003	1.005	
Cause (Food > Climate) *									
Binding Moral Foundations	1.02	0.02	0.98	1.05	0.92	0.36	-0.009	0.959	

Note. OR: odds ratio, SE: standard error of the mean, CI: confidence interval. Columns up to *p* duplicate those from Table S10 with rows that do not correspond to interaction effects not shown for brevity. The final two columns give the results of the simulation analyses. Effect 95% corresponds to the 95% percentile of the distribution of raw differences in *Ks* for climate vs. food in 1,000 simulated datasets. The null hypothesis here is 0 and the effect in the real data is positive. Interventions with a significant effect are shown in bold italic and none of the 95th percentile values cross 0. Model 95% provides the 95% percentile of the distribution in odds ratios for the mixed model of choices in 1,000 simulated datasets. The null hypothesis here is 1 and the effect in the real data is above 1 (see first OR column). None of the interventions with a significant effect cross 1.

It is also not clear to me why negative emotions and pluralistic ignorance ‘flip places’ between the first and second analysis. Specifically, negative emotions are reported as having a significant effect in the first analysis and not the second, and vice versa for pluralistic ignorance. Since these analyses are focussing on the same research question, I would expect the results to be similar across them.

Response: Thank you for this detailed consideration of our results. As can be seen from Figure 3 (choices) and Figure 4 (*K* parameters), the pattern of results and relative effect sizes is very similar across measures. While analyses of these two measures are part of addressing the same research question, they are not identical and some variation in results is to be expected. The choices are the raw proportion of times participants chose the work offer which has strengths in that is easy to interpret and understand for any reader. The *K* values separate the effort discounting aspect of choices from decision consistency, both of which contribute to how people make decisions, which is a key strength of the computational modelling approach. We take a stringent approach to reporting an intervention as having a significant effect, only if it is significant across our four mixed models (main and three control variables). This also contributes to the inconsistency you mention, for example Pluralistic Ignorance is significant in the main and one control analysis but just over the threshold in two control analyses. We think this stringent testing is important, including for the points raised above about ensuring results are robust. We would be happy to mention this point in the manuscript if needed.

Lastly, it is not clear to me why the authors focus on the exerted effort relative to the food condition as the main outcome of interest. I understand that this offers a relevant control, but the main effect on overall exerted effort in the climate condition seems like the primary

outcome to me. The authors do analyze the overall exerted effort as an outcome but do not consider the effect of the interventions here. I do not follow the rationale here since the effect of the interventions is the main topic of this paper.

***Response:** We appreciate the opportunity to clarify this aspect. We focus on the relative exerted effort between causes as we know that people vary substantially in their overall willingness to exert effort (also demonstrated in Figure 5c & d through associations with apathy and subjective effort). We want to consider their pro-environmental motivation specifically and this is particularly important for testing the effect of the interventions as these were manipulated between-subject. The only way to do this using overall exerted effort for the climate as the measure of interest would be to test participants before and after the intervention, we have added this as a suggestion for future work in our discussion:*

“Designs that measure motivation on the PEET pre- and post-intervention are needed to capture intervention effects within participant and could then use overall climate motivation as the main outcome, rather than relative to the food charity.”

Our mixed effects models have terms for the overall effect of effort and reward across conditions because they also have a subject-specific intercept that accounts for overall motivation. We only report findings, such as the negative effect of effort on overall exerted effort, when they apply both in the food and climate cause, without significant differences in these effects. We have clarified this in the manuscript:

“We used generalized linear mixed-effects models (GLMMs) to determine whether choices between working or resting were sensitive to effort and reward and additional follow-up tests to establish that these effects were found for both climate and food separately, as well as when combined.”